# HART: Human Aligned Reconstruction Transformer

## Abstract

We introduce HART, a unified framework for sparse-view human reconstruction. Given a small set of uncalibrated RGB images of a person as input, it outputs a watertight clothed mesh, the aligned SMPL-X body mesh, and a Gaussian-splat representation for photorealistic novel-view rendering. Prior methods for clothed human reconstruction either optimize parametric templates, which overlook loose garments and human-object interactions, or train implicit functions under simplified camera assumptions, limiting applicability in real scenes. In contrast, HART predicts per-pixel 3D point maps, normals, and body correspondences, and employs an occlusion-aware Poisson reconstruction to recover complete geometry, even in self-occluded regions. These predictions also align with a parametric SMPL-X body model, ensuring that reconstructed geometry remains consistent with human structure while capturing loose clothing and interactions. These human-aligned meshes initialize Gaussian splats to further enable sparse-view rendering. While trained on only 2.3K synthetic scans, HART achieves state-of-the-art results: Chamfer Distance improves by **18–23%** for clothed-mesh reconstruction, PA-V2V drops by **6–27%** for SMPL-X estimation, LPIPS decreases by **15–27%** for novel-view synthesis on a wide range of datasets. These results suggest that feed-forward transformers offer a scalable, data-efficient paradigm for robust human reconstruction and naturally improving as more training data becomes available. Code and models will be released.

## 1 Introduction

3D human reconstruction is crucial for applications like virtual try-on, AR/VR, telepresence, and digital content creation. Recent methods based on NeRF (Peng et al., 2021a; Guo et al., 2023; Wang et al., 2022) and 3D Gaussian Splatting (3DGS) (Qian et al., 2024; Guo et al., 2025; Li et al., 2024b) excel in both rendering and geometry reconstruction. However, they either require dense-view inputs, accurate camera calibrations, or robust SMPL (Loper et al., 2015) estimations, while training such models for a single person could take minutes to hours. In a more practical scenario, feed-forward inference from a set of unposed sparse-view human images would be preferable due to efficiency and scalability, yet accurately inferring geometry and appearance from such limited inputs remains challenging due to the complexity of human bodies (e.g. articulations and self-occlusions).

Earlier works have tackled the problem of sparse-view human geometry reconstruction by learning generalizable pixel-aligned implicit functions (Saito et al., 2019; 2020; Cao et al., 2023), achieving direct clothed human mesh regression from sparse-view RGB images. However, such methods often assume orthographic projections, which significantly limits their generalization ability to real-world perspective images. Recently, advances in Score Distillation Sampling (SDS) (Poole et al., 2022) have enabled human geometry distillation from pretrained diffusion models, achieving detailed surface reconstruction from uncalibrated images (Xiu et al., 2024; Zeng et al., 2023). However, they optimize human poses in canonical SMPL poses, which often fail to recover complete geometry for loose garments and human-object interactions.

In the broader 3D reconstruction community, general-purpose feed-forward approaches have made rapid progress. Recent works in transformer-based backbones (Wang et al., 2024b; Yang et al., 2025; Zhang et al., 2025; Wang et al., 2025; Tang et al., 2024b) have significantly advanced calibration-free 3D reconstruction from sparse views, enabling a wide range of downstream 3D vision tasks, such as camera pose regression, tracking, and novel view synthesis, with impressive generalization abilities to real-world images. These approaches form the natural backbone for our method. However, they only output raw point clouds that require further meshing, and their predictions remain limited to pixels visible in the input images. As a result, they fail to capture occluded regions unseen in the input images – especially problematic in human reconstruction with pervasive self-occlusion.

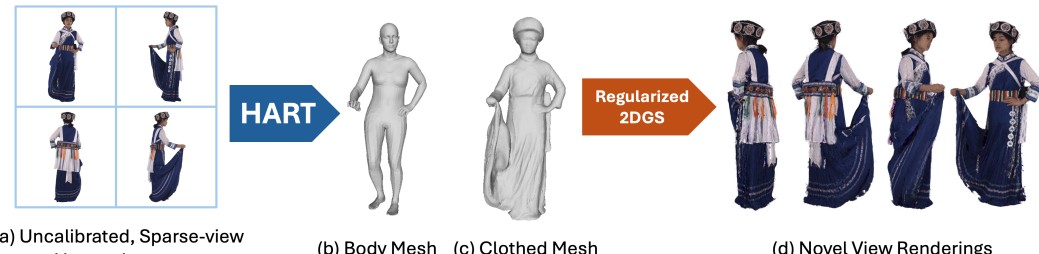

(a) Uncalibrated, Sparse-view Human Images  (b) Body Mesh  (c) Clothed Mesh  (d) Novel View Renderings

Figure 1: Given **(a)** uncalibrated, sparse-view human images, our method HART is a unified framework that simultaneously reconstructs **(b)** the underlying SMPL-X body mesh and **(c)** the clothed mesh. **(d)** Our clothed mesh prediction serves as an initialization and regularization to further enable novel view synthesis from sparse views.

Beyond clothed geometry, estimating a parametric body mesh from multi-view inputs is also of high interest. Existing approaches typically rely on keypoint-based fitting (Pavlakos et al., 2019; eas, 2021; Shuai et al., 2022), which can be brittle under complex poses, self-occlusions, and loose garments. In contrast, our dense point map predictions naturally serve as strong geometric priors. By augmenting our transformer backbone with per-pixel SMPL-X (Pavlakos et al., 2019) tightness (Li et al., 2025) and body-part label heads, we enable prediction of accurate SMPL-X parameters alongside clothed meshes.

We additionally find that our high-quality clothed mesh reconstruction can serve as a good proxy for novel view synthesis. By initializing Gaussian surfels (Huang et al., 2024) from our predicted mesh faces, we could achieve sparse-view human rendering via regularized 2D Gaussian Splatting (Guédon et al., 2025). Our key observation is that *constraining Gaussian Splatting with accurate clothed geometry substantially improves rendering quality while mitigating overfitting*.

In summary, our key contributions lie in unifying feed-forward point map prediction with novel geometry completion modules and parametric body estimation for robust human reconstruction and rendering. Specifically, we introduce Human Aligned Reconstruction Transformer (HART), a unified transformer-based architecture that jointly predicts point maps, surface normals, and SMPL-X tightness vectors with semantic body-part labels. This design enables *simultaneous reconstruction of detailed clothed meshes* and the underlying SMPL-X body meshes in a feed-forward manner. To overcome the limitations of point-map–based frameworks in handling self-occlusions, we introduce a **3D U-Net in the Differentiable Poisson Surface Reconstruction (DPSR)** module. By refining the indicator grid with residual corrections, HART *recovers complete and watertight clothed geometry*. While trained on only $2.3K$ human scans, our method achieves state-of-the-art performance across multiple benchmarks, including clothed mesh reconstruction, sparse-view SMPL-X estimation, and novel view synthesis. Extensive quantitative and qualitative evaluations further demonstrate that HART generalizes well to real-world human images with loose garments.

## 2 RELATED WORK

**Structure from Motion**  Structure from Motion (SfM) is a fundamental computer vision problem that involves estimating camera parameters and reconstructing sparse 3D point clouds from multiple images of a static scene (Hartley & Zisserman, 2000; Oliensis, 2000; Özyeşil et al., 2017), with COLMAP (Schönberger & Frahm, 2016) being the most widely adopted framework. Recent years have seen significant advances through deep learning integration, improving keypoint detection (DeTone et al., 2018; Dusmanu et al., 2019; Tyszkiewicz et al., 2020; Yi et al., 2016) and image matching (Chen et al., 2021; Lindenberger et al., 2023; Sun et al., 2021), culminating in end-to-end differentiable SfM approaches (Wang et al., 2024a). A paradigm shift emerged with DUSt3R (Wang et al., 2024b) and MASt3R (Leroy et al., 2024), which directly estimate aligned dense point maps from image pairs without requiring camera parameters, and produce these parameters along with 3D reconstructions. Most recently, VGGT (Wang et al., 2025) demonstrates that a standard transformer trained on extensive 3D data can directly predict all 3D attributes (cameras, depth, point maps, tracks) in a single feed-forward pass, achieving state-of-the-art results without post-processing optimization. Our work adopts VGGT into the human reconstruction domain and goes beyond point map reconstruction by simultaneously estimating a detailed human mesh and underlying SMPL-X meshes.

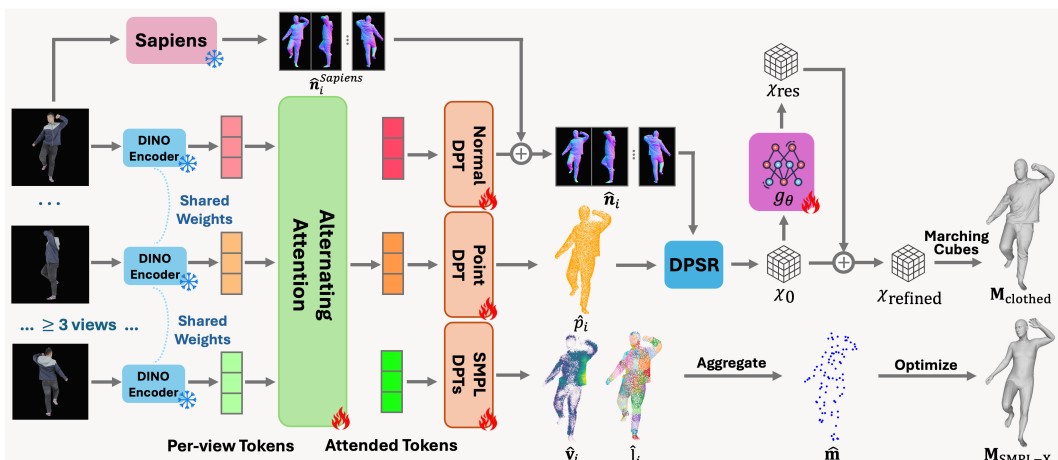

Figure 2: **Overview of our Network Architecture.** Given $N$ uncalibrated human images, our HART transformer first maps input images $\{I_i\}_{i=1}^{N}$ into per-pixel point maps $\hat{p}_i$, refined normal maps $\hat{n}_i$, SMPL-X tightness vectors $\hat{v}_i$ and body part labels $\hat{l}_i$. The oriented point maps $\hat{p}_i, \hat{n}_i$ for all views are merged and converted to an indicator grid $\chi_{\text{refined}}$ via Differentiable Poisson Surface Reconstruction (DPSR). A 3D-UNet $g_\theta$ is used for grid refinement to account for self-occlusions and a clothed mesh reconstruction $\mathbf{M}_{\text{clothed}}$ can be obtained by running marching cubes. The SMPL-X tightness vectors and label maps are aggregated into body markers $\hat{\mathbf{m}}$ out of which we could optimize a SMPL-X mesh $\mathbf{M}_{\text{SMPL-X}}$.

**Sparse-view 3D Reconstruction**  Neural Radiance Fields (NeRF) (Mildenhall et al., 2020) have revolutionized novel view synthesis and 3D reconstruction from multi-view images, while 3D Gaussian Splatting (3DGS) (Kerbl et al., 2023) has made radiance field learning and rendering significantly more efficient and scalable. However, the vanilla NeRF/3DGS models require costly per-scene optimization and large numbers of input images (typically 20-100 views) to achieve high-quality. Recent works have explored learning-based approaches that directly reconstruct NeRF or 3DGS from sparse-view images in a feed-forward manner. (Suhail et al., 2022; Lin et al., 2022; Xu et al., 2024; Wu et al., 2024; Hong et al., 2024) proposes to predict radiance fields from sparse-view images using either neural networks trained on large-scale multi-view datasets. (Zhang et al., 2024; Tang et al., 2024a; Chen et al., 2024b; Xu et al., 2025) predicts per-pixel Gaussian splats instead of implicit NeRF, enabling real-time rendering and better scalability to high-resolution images. (Chen et al., 2024a) introduces a latent voxel grid representation to encode 3D Gaussians, achieving better 3D consistency in wide baseline settings. These methods demonstrate promising results on general 3D scenes, but typically require calibrated camera poses as inputs.

**Human Reconstruction from Sparse-view Images**  Earlier works such as (Saito et al., 2019; 2020; Huang et al., 2020) pioneered the use of neural fields for high-fidelity 3D human reconstruction from single RGB images. (Xiu et al., 2022; Zheng et al., 2021) further improved the reconstruction quality by leveraging parametric human models like SMPL (Loper et al., 2015) as guidance. Another line of work (Peng et al., 2021a; Guo et al., 2023; Weng et al., 2022; Wang et al., 2022; Qian et al., 2024) combines NeRF/3DGS and human models for human reconstruction from sparse-view or even monocular videos. These methods usually rely on per-scene optimization over videos. Other works (Cao et al., 2023; Yu et al., 2025; Zhou et al., 2025a; Kwon et al., 2024; Hu et al., 2024) try to directly predict human reconstruction from sparse-view images (typically 3-8 views) in a feed-forward manner. Our work also falls into this category. We distinguish our approach from prior works by leveraging recent point-map-based reconstruction models to process *calibration-free* human images. In contrast, existing methods either rely on accurate camera parameters or assume orthographic projections, an assumption that holds only for synthetic data and fails on real-world images.

## 3 METHOD

We begin by detailing the architecture of our transformer with per-pixel prediction heads in Sec. 3.1. Sec. 3.2 presents the subsequent occlusion-aware DPSR module for complete human surface re-

construction. Sec. 3.3 outlines our training details, and finally, our geometry-informed novel view synthesis pipeline is described in Sec. 3.4.

## 3.1 HART: Human Aligned Reconstruction Transformer

At the core of our method is a human-aligned transformer with downstream heads for per-pixel human attribute predictions. Given a set of $N$ ($N \geq 3$) uncalibrated human images $\{I_i \in \mathbb{R}^{3 \times H \times W}\}_{i=1}^{N}$ captured in the same body pose, the transformer $f$ is a function that maps the images into a set of per-pixel attributes:

$$f(\{I_i\}_{i=1}^{N}) = \{\hat{p}_i, \hat{\mathbf{n}}_i, \hat{\mathbf{v}}_i, \hat{l}_i\}_{i=1}^{N}, \tag{1}$$

where $\hat{p}_i, \hat{\mathbf{n}}_i, \hat{\mathbf{v}}_i \in \mathbb{R}^{3 \times H \times W}$, and $\hat{l}_i \in \mathbb{N}^{H \times W}$ denote the predicted point map, normal map, SMPL-X tightness map, and SMPL-X body-part label map for input image $I_i$. The oriented point predictions $\hat{p}_i$ and $\hat{\mathbf{n}}_i$ are used to reconstruct the clothed mesh $\mathbf{M}_{\text{clothed}}$, while the SMPL-X tightness and label maps guide the estimation of the parametric body mesh $\mathbf{M}_{\text{SMPL-X}}$. An overview of our network architecture is shown in Fig. 2.

We adopt VGGT (Wang et al., 2025), a recent state-of-the-art feed-forward transformer for general-purpose 3D reconstruction, as the backbone of our framework. Each input image $I_i$ is first patchified into $K$ tokens, denoted as $t^{I_i} \in \mathbb{R}^{K \times C}$, using the DINOv2 encoder (Oquab et al., 2023). These per-view tokens are then fused across images using the alternating attention layers from VGGT, which allows the network to capture both intra-view spatial relationships and cross-view geometric correspondences, forming a powerful representation for subsequent prediction heads.

After the attention layers, the fused tokens $\hat{t}^{I_i}$ for each image $I_i$ are transformed to dense per-pixel downstream feature maps $F_i \in \mathbb{R}^{C \times H \times W}$ via prediction heads. Following (Wang et al., 2024b; 2025), we adopt DPT head (Ranftl et al., 2021) as our prediction heads.

### 3.1.1 Point Head and Camera Pose Optimization

Similar to (Wang et al., 2024b; 2025), our predicted point maps are *viewpoint-invariant*, meaning that these 3D points are expressed in the coordinate system of the first camera, which we designate as the world reference frame.

Our point map loss follows the formulation of (Wang et al., 2024b) with an aleatoric uncertainty (Kendall & Cipolla, 2016; Novotny et al., 2018) term. Given the ground-truth point map $p_i$ and predicted confidence map $\hat{C}_{p_i}$, the loss is defined as:

$$\mathcal{L}_{\text{point}} = \sum_{i=1}^{N} ||\hat{C}_{p_i} \odot (\hat{p}_i - p_i)||_1 - \alpha \log \hat{C}_{p_i}, \tag{2}$$

It is worth noting that, unlike (Wang et al., 2024b; 2025) and other common SfM frameworks, we do not assume the camera's principal point at the image center. We observe that this assumption significantly restricts generalization when working with foreground-focused human images, and thus, we explicitly relax it in our formulation. Therefore, we do not adopt VGGT's pretrained camera head, which assumes centered principal points. Instead, we use RANSAC (Fischler & Bolles, 1981) and PnP (Lepetit et al., 2009) as in (Wang et al., 2024b; Yang et al., 2025) to estimate camera parameters from predicted point maps.

### 3.1.2 Residual Normal Head

Accurate surface normals are critical for high-fidelity surface reconstruction. We find that directly predicting normals using a DPT head often results in overly smooth or blurry estimates, likely due to the limited capacity of the VGGT backbone for fine-grained local geometry.

To overcome this, we adopt a residual-learning strategy. Instead of learning full normals from scratch, the network predicts *residual normals* $\mathbf{n}_i^{\text{res}} \in \mathbb{R}^{3 \times H \times W}$ with respect to the results of a state-of-the-art human normal estimator (Khirodkar et al., 2024), denoted $\mathbf{n}_i^{\text{Sapiens}}$. The final normal map $\mathbf{n}_i$ is computed as:

$$\hat{\mathbf{n}}_i = \text{normalize}\left(\hat{\mathbf{n}}_i^{\text{Sapiens}} + \hat{\mathbf{n}}_i^{\text{res}}\right), \tag{3}$$

where $\text{normalize}(\cdot)$ enforces unit length.

This residual formulation leverages the strong prior from $\hat{\mathbf{n}}_i^{\mathrm{Sapiens}}$ while refining high-frequency details and enforcing multi-view consistency; by integrating independently predicted monocular normals across views, our residual normal head yields more coherent and detailed results, which in turn significantly improve subsequent surface reconstruction.

Similar to our point map loss, we define the normal map loss as:

$$\mathcal{L}_{\mathrm{normal}} = \sum_{i=1}^{N}(\hat{C}_{\mathbf{n}_i} \odot (1 - \hat{\mathbf{n}}_i \cdot \mathbf{n}_i)) - \alpha\log\hat{C}_{\mathbf{n}_i}, \tag{4}$$

Finally, we leverage camera rotation matrices $R_{c2w}$ estimated from our predicted point maps to transform normals to the world reference frame: $\hat{\mathbf{n}}_i^{\mathrm{world}} = R_{c2w}\hat{\mathbf{n}}_i$.

### 3.1.3 SMPL-X HEADS

We predict two key components that establish correspondence between the clothed surface and the underlying SMPL-X body: **tightness vectors** and **body part labels**. These predictions will enable us to fit SMPL-X meshes to clothed humans.

**Tightness Vector Heads.** Following ETCH (Li et al., 2025), we predict tightness vectors $\hat{\mathbf{v}}_i$ that point from clothed surface points to their corresponding locations on the underlying body surface. Each tightness vector is decomposed into direction $\hat{\mathbf{d}}_i$ and magnitude $\hat{b}_i$ components, where $\hat{\mathbf{v}}_i = \hat{b}_i\hat{\mathbf{d}}_i$. The direction component captures the geometric relationship between clothing and the body, while the magnitude reflects the looseness of the clothing and varies with the garment type and body region. Contrary to ETCH, which takes sparse point clouds as inputs, we directly predict per-pixel tightness vectors from images using two individual DPT heads for tightness directions and magnitudes. This results in much denser tightness predictions and thus better body fitting results.

**Body Part Label Head.** We include another DPT head that predicts body part assignment map $\hat{l}_i$ with corresponding confidence map $\hat{c}_i$, mapping each clothed surface point to one of 86 predefined SMPL-X body markers. This semantic labeling enables us to aggregate tightness-corrected points into sparse body markers, providing anchors for SMPL-X parameter estimation.

**Marker Aggregation and SMPL-X Fitting.** Given the predicted tightness vectors and part labels, we first compute inner body points as $\hat{\mathbf{y}}_i = \hat{p}_i + \hat{\mathbf{v}}_i$, where $\hat{p}_i$ are the clothed surface points from the Point Head. We then aggregate points with the same part label into sparse body markers:

$$\hat{\mathbf{m}}_k = \frac{\sum_{i:\hat{l}_i=k}(\hat{c}_i)^\alpha\hat{\mathbf{y}}_i}{\sum_{i:\hat{l}_i=k}(\hat{c}_i)^\alpha} \tag{5}$$

where $\hat{\mathbf{m}}_k$ represents the $k$-th body marker. $\alpha$ is a hyperparameter that controls the influence of confidence weights.

We optimize SMPL-X parameters $(\boldsymbol{\theta}, \boldsymbol{\beta}, \mathbf{t})$ along with a scaling factor $s$ to get the final SMPL-X mesh $\mathbf{M}_{\mathrm{SMPL\text{-}X}}$, by minimizing the L2 distance between predicted markers and corresponding SMPL-X surface points:

$$\min_{s,\boldsymbol{\theta},\boldsymbol{\beta},\mathbf{t}} \sum_{k=1}^{86} \|\tilde{\mathbf{m}}_k - \hat{\mathbf{m}}_k\|^2 + \lambda_{reg}\mathcal{L}_{reg}, \tag{6}$$

where $\tilde{\mathbf{m}}_k$ are markers on the current SMPL-X estimate, and $\mathcal{L}_{reg}$ is an L2 regularization of $\boldsymbol{\theta}$ and $\boldsymbol{\beta}$. This formulation transforms the challenging clothed human fitting problem into a well-posed sparse marker fitting task, enabling robust and efficient body parameter estimation even under loose clothing. Instead of directly optimizing SMPL-X poses $\boldsymbol{\theta}$, we optimize the pose embedding of VPoser (Pavlakos et al., 2019), which provides a stronger pose regularization.

Following ETCH, the training loss of our SMPL-X branch is a combination of losses on tightness direction, magnitude, label classification, and confidence, defined as:

$$\mathcal{L}_{\mathrm{SMPL}} = \mathcal{L}_d + \mathcal{L}_b + \mathcal{L}_l + \mathcal{L}_c,$$

$$\mathcal{L}_d = \sum_{i=1}^{N}\left(1 - \hat{\mathbf{d}}_i \cdot \mathbf{d}_i\right), \quad \mathcal{L}_b = \sum_{i=1}^{N}(\hat{b}_i - b_i)^2, \quad \mathcal{L}_l = -\frac{1}{N}\log(p_{i,k=\hat{l}_i}), \quad \mathcal{L}_c = \sum_{i=1}^{N}(\hat{c}_i - c_i)^2, \tag{7}$$

where $\mathbf{d}_i$, $b_i$, $l_i$, $c_i$ represent ground-truth tightness vector direction, magnitude, part label, and geodesic distance-based confidence, respectively. For more technical details about our SMPL-X heads, please refer to the appendix.

### 3.2 Clothed Mesh Reconstruction via Occlusion-aware DPSR

With predicted point maps and normal maps, one could apply classical Poisson surface reconstruction methods (Kazhdan & Hoppe, 2013) to obtain a mesh of the human. To this end, our clothed-mesh reconstruction branch builds on the *Differentiable Poisson surface reconstruction* (DPSR) framework of SAP (Peng et al., 2021b) to reconstruct the indicator grid of human shapes, followed by a refinement network to handle occluded regions not seen in input images. This enables our framework to directly produce a watertight mesh $\mathbf{M}_{\text{clothed}}$ via marching cubes (Lorensen & Cline, 1987).

**Initial Indicator Grid Generation.** Given the predicted human point maps $\mathbf{P} = \{\hat{p}_i[M_i] \in \mathbb{R}^3\}_{i=1}^N$ and world-space normals $\mathbf{N} = \{\hat{\mathbf{n}}_i^{\text{world}}[M_i] \in \mathbb{R}^3\}_{i=1}^N$ ($M_i$ represents foreground human masks), we apply DPSR to generate an initial indicator grid $\chi_0 \in \mathbb{R}^{r \times r \times r}$. The DPSR solves the Poisson equation $\nabla^2 \chi = \nabla \cdot \mathbf{v}$, where $\mathbf{v}$ represents the normal vector field rasterized from the oriented point cloud $(\mathbf{P}, \mathbf{N})$. We refer readers to (Peng et al., 2021b) for implementation details.

**3D-UNet Refinement.** The initial indicator grid $\chi_0$, while geometrically consistent, often suffers from missing details and gaps in unobserved regions due to the sparse and potentially incomplete nature of the input point maps. This motivates us to learn a 3D-UNet to refine the initial reconstructed indicator grid $\chi_0$.

Specifically, the 3D-UNet $g_\theta$ takes the coarse indicator grid $\chi_0 \in \mathbb{R}^{r \times r \times r}$ as input, and predicts a residual indicator grid $\chi_{\text{res}}$ with the same resolution. The refined indicator grid is obtained via $\chi_{\text{refined}} = \chi_0 + \chi_{\text{res}}$. For the detailed architecture of this module, please refer to the appendix.

We supervise the final refined indicator grid via $\chi_{\text{gt}}$ obtained from ground-truth mesh:

$$\mathcal{L}_{\text{DPSR}} = \frac{1}{r^3} \sum_x \left( \chi_{\text{refined}}(x) - \chi_{\text{gt}}(x) \right)^2. \tag{8}$$

### 3.3 Training

**Training Objective.** The total loss of our HART transformer is defined as

$$\mathcal{L} = \mathcal{L}_{\text{point}} + \mathcal{L}_{\text{normal}} + \mathcal{L}_{\text{DPSR}} + \mathcal{L}_{\text{SMPL}}. \tag{9}$$

Similar to VGGT, we observe that our framework converges stably without the need to weight individual loss terms against each other. Please refer to the appendix for implementation details.

**Training Data.** We train our network with 2,345 subjects from the THuman 2.1 (Yu et al., 2021) dataset with textured scans and ground-truth SMPL-X annotations. We render each subject into 96 views in a 360-degree azimuth trajectory and apply center-cropping around the center of the human masks to only focus on the foreground regions of the human images.

### 3.4 Geometry-Informed Novel View Synthesis

Our accurate clothed mesh reconstruction also enables high-quality novel view synthesis (NVS) from sparse-view inputs. Inspired by a recent geometrically-regularized Gaussian splatting method (Guédon et al., 2025), we initialize 2D Gaussian surfels directly on our reconstructed mesh $\mathbf{M}_{\text{clothed}}$ and optimize their attributes to best fit the input images.

**Gaussian Surfel Initialization.** We instantiate 2D Gaussians at the face centers of our reconstructed mesh, while their orientations are aligned with the local surface normals. Following (Guédon et al., 2025), we parameterize each Gaussian's covariance matrix to lie tangent to the surface, forming 2D surfels that faithfully respect the underlying geometry.

**Optimization.** We generally follow (Guédon et al., 2025) but with some modifications: We find it beneficial to disable Gaussian densification and pruning, and fix the number of Gaussians to the number of mesh faces, as they are already sufficiently dense and accurate. We also find it effective to apply a lower learning rate to Gaussian means, scales, and rotations, which further stabilizes training.

We optimize the Gaussian parameters using a combination of losses:

$$\mathcal{L}_{\text{rendering}} = \mathcal{L}_{\text{photo}} + \lambda_d \mathcal{L}_d + \lambda_n \mathcal{L}_n + \lambda_{struct} \mathcal{L}_{struct}, \tag{10}$$

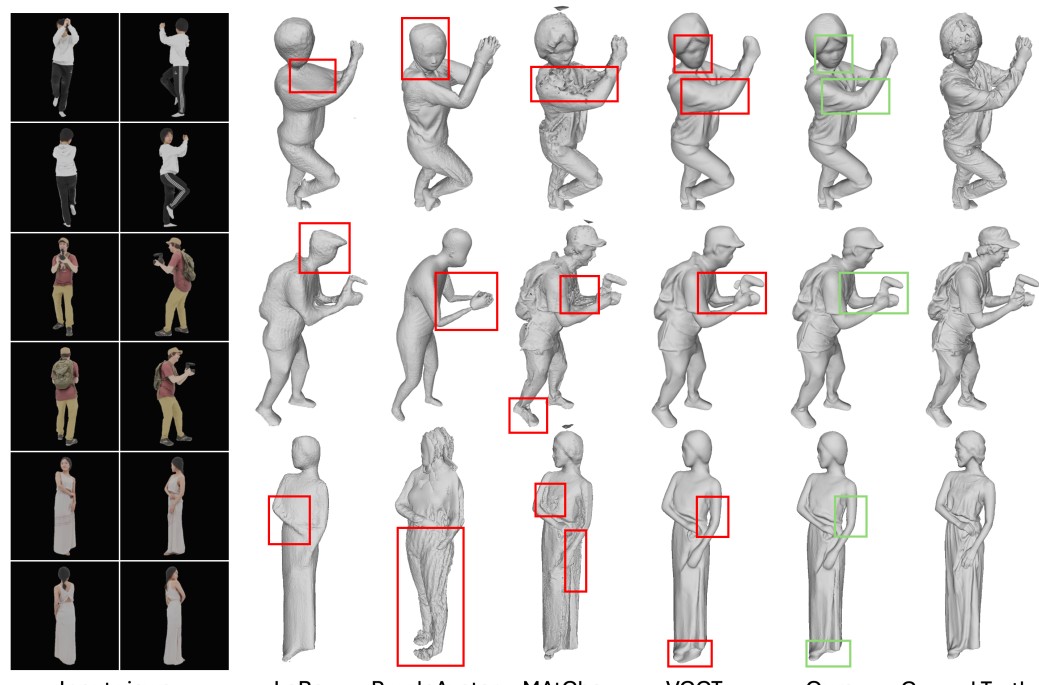

Input views    LaRa    PuzzleAvatar    MAtCha    VGGT    Ours    Ground Truth

Figure 3: **Clothed Mesh Reconstruction from 4 views.** We show 1 subject from THuman 2.1 (row 1) and 2 from 2K2K test sets (rows 2–3). In contrast to various baselines, our method can recover detailed geometry in both observed and occluded regions.

where $\mathcal{L}_{\text{photo}}$, $\mathcal{L}_d$, and $\mathcal{L}_n$ denote the photometric loss, the depth distortion, and normal consistency regularization losses adopted from 2DGS (Huang et al., 2024). The structure loss $\mathcal{L}_{\text{struct}}$ follows (Guédon et al., 2025) and regularizes the Gaussian geometry with our reconstructed mesh.

## 4 EXPERIMENTS

We evaluate our method on clothed mesh reconstruction, SMPL-X estimation, and novel view synthesis. All comparisons with baselines are conducted under a fixed setting of 4 input views. In the appendix, we provide additional results under varying numbers of views as well as baseline details/hyperparameters and ablation studies.

### 4.1 DATASET

We use three test datasets as our major testbeds: 1) the THuman 2.1 test set with 100 subjects for in-domain evaluation of all three tasks. 2) A subset of 100 subjects from the 2K2K dataset (Han et al., 2023) for cross-domain mesh reconstruction and SMPL-X estimation evaluation. This dataset has more diversity in age, clothing styles, and human–object interactions not present in our training set. 3) The DNA-Rendering dataset (Cheng et al., 2023) for cross-domain novel-view synthesis evaluation; the dataset contains dense-view real-world *raw images* of 41 subjects wearing loose garments and performing intricate human-object interactions.

### 4.2 CLOTHED MESH RECONSTRUCTION

**Baselines.** We adopt VGGT (Wang et al., 2025), MAtCha (Guédon et al., 2025), Puzzle Avatar (Xiu et al., 2024), and LaRa (Chen et al., 2024a) as baselines. We finetune VGGT and LaRA on the same training set as our method. For MAtCha, we replace MASt3R-SfM (Leroy et al., 2024) with our estimated camera, while also using Sapiens (Khirodkar et al., 2024) for depth estimation, as Sapiens is specialized in the human domain.

**Metrics.** We evaluate clothed mesh reconstruction quality using Chamfer Distance (CD) ($\times 10^{-3}$), F-Score at a threshold of $0.5\%$, and Normal Consistency (NC). For Chamfer Distance, we additionally report its two directional components: *Accuracy*, defined as the mean distance from each predicted point to its closest ground-truth point, and *Completeness*, defined as the mean distance

Table 1: **Quantitative Comparison of Clothed Mesh Reconstruction.** Our method achieves the best performance across nearly all metrics, demonstrating both high-fidelity in-domain reconstruction and strong cross-domain generalization.

| Methods | THuman 2.1 (In-domain) | | | | | 2K2K (Cross-domain) | | | | |
|---|---|---|---|---|---|---|---|---|---|---|
| | Acc. | Comp. | CD | F-Score | NC | Acc. | Comp. | CD | F-Score | NC |
| VGGT (Wang et al., 2025) | 0.0070 | 0.0140 | 0.0209 | 0.9285 | / | **0.0072** | 0.0151 | 0.0222 | 0.9274 | / |
| MAtCha (Guédon et al., 2025) | 0.1264 | 0.0161 | 0.1425 | 0.6793 | 0.6506 | 0.1175 | 0.0138 | 0.1313 | 0.6938 | 0.6956 |
| Puzzle Avatar (Xiu et al., 2024) | 0.1311 | 0.1652 | 0.2963 | 0.3916 | 0.7255 | 0.1374 | 0.1916 | 0.3291 | 0.4095 | 0.7587 |
| LaRa (Chen et al., 2024a) | 0.0334 | 0.0466 | 0.0800 | 0.6466 | 0.8257 | 0.0279 | 0.0409 | 0.0688 | 0.6645 | 0.8705 |
| Ours | **0.0067** | **0.0105** | **0.0172** | **0.9354** | **0.9125** | 0.0077 | **0.0093** | **0.0170** | **0.9301** | **0.9479** |

Table 2: **Quantitative Comparisons of Sparse-view SMPL-X estimation.** Across both in-domain and cross-domain test sets, ours consistently reconstructs more accurate body meshes than others.

| Methods | THuman 2.1 (In-domain) | | 2K2K (Cross-domain) | |
|---|---|---|---|---|
| | PA-V2V | PA-MPJPE | PA-V2V | PA-MPJPE |
| MV-SMPLify-X (Zheng et al., 2021; Pavlakos et al., 2019) | 21.66 | 26.11 | 24.77 | 29.81 |
| EasyMocap (eas, 2021; Shuai et al., 2022) | 25.89 | 31.22 | 24.36 | 26.22 |
| ETCH (Li et al., 2025) | 21.49 | 22.87 | 27.06 | 26.22 |
| Ours | **15.72** | **16.18** | **22.86** | **24.49** |

from each ground-truth point to its closest prediction. The total Chamfer Distance is reported as the sum of Accuracy and Completeness.

**Discussion.** Fig. 3 and Tab. 1 present qualitative and quantitative comparisons for clothed mesh reconstruction. LaRa yields overly smooth surfaces. PuzzleAvatar, constrained by its reliance on parametric body templates, produces inaccurate body shapes and fails to capture loose garments or object interactions. MAtCha recovers overall shapes but introduces noisy surfaces. The most competitive baseline, VGGT, produces point maps that could be converted to reasonable meshes with Poisson surface reconstructions. However, it struggles with self-occluded regions. In contrast, our method better captures occluded areas and adds fine details (e.g., facial regions), thanks to our residual 3D-UNet and normal predictions. These differences are also reflected in the quantitative results, as we outperform VGGT significantly w.r.t. completeness and normal consistency.

## 4.3 SMPL-X ESTIMATION

**Baselines.** We compare our approach against three baselines: EasyMocap (eas, 2021; Shuai et al., 2022), multi-view variants of SMPLify-X (MV-SMPLify-X) (Pavlakos et al., 2019; Zheng et al., 2021), and ETCH (Li et al., 2025). We use (Xu et al., 2022) for 2D keypoint detection required by EasyMoCap and MV-SMPLify-X, while finetuning ETCH on our clothed mesh reconstructions on the THuman 2.1 training set.

**Metrics.** We evaluate Mean Vertex-to-Vertex Error (PA-V2V) by comparing all vertices of the SMPL-X mesh, and Mean Per-Joint Position Error (PA-MPJPE) by comparing the body joints. Both metrics are computed after Procrustes Alignment (Gower, 1975) and are reported in millimeters.

**Discussion.** Fig. 4 and Tab. 2 present qualitative and quantitative comparisons on SMPL-X estimation. EasyMocap and MV-SMPLify-X often yield meshes with inaccurate head poses and body shapes, while ETCH struggles with fine details such as hands and feet as it only allows a small number of input 3D points (around 5000). In contrast, our method produces more reliable SMPL-X

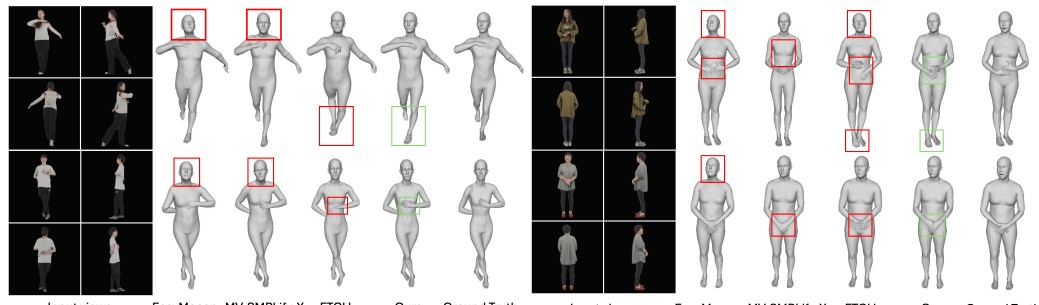

Input views   EasyMocap   MV-SMPLify-X   ETCH   Ours   Ground Truth          Input views   EasyMocap   MV-SMPLify-X   ETCH   Ours   Ground Truth

Figure 4: **SMPL-X Mesh Reconstruction from 4 Views:** 2 subjects from THuman (left) and 2 from 2K2K test sets (right). Keypoint-based EasyMocap and MV-SMPLify-X produce inaccurate head poses and body shapes, while ETCH often misstitches reconstructed feet/hands.

Table 3: **Quantitative Comparison of Novel View Synthesis.** Ours consistently outperforms prior arts across synthetic (THuman 2.1) and real-world (DNA Rendering) test sets – with higher fidelity renderings, better perceptual quality (SSIM, LPIPS), and competitive realism (FID).

| Methods | THuman 2.1 (Synthetic) | | | | DNA Rendering (Real World) | | | |
|---|---|---|---|---|---|---|---|---|
| | PSNR | SSIM | LPIPS | FID | PSNR | SSIM | LPIPS | FID |
| LaRa (Chen et al., 2024a) | 29.05 | 0.9464 | 0.0935 | 68.12 | 26.71 | 0.9209 | 0.1093 | 98.09 |
| SEVA (Zhou et al., 2025b) | 21.65 | 0.7843 | 0.0909 | **5.03** | 21.67 | 0.8029 | 0.1075 | **29.68** |
| MAtCha (Guédon et al., 2025) | 30.44 | 0.9546 | 0.0537 | 21.76 | 26.77 | 0.9214 | 0.0708 | 40.77 |
| Ours | **31.70** | **0.9675** | **0.0390** | 14.24 | **27.54** | **0.9349** | **0.0600** | 36.29 |

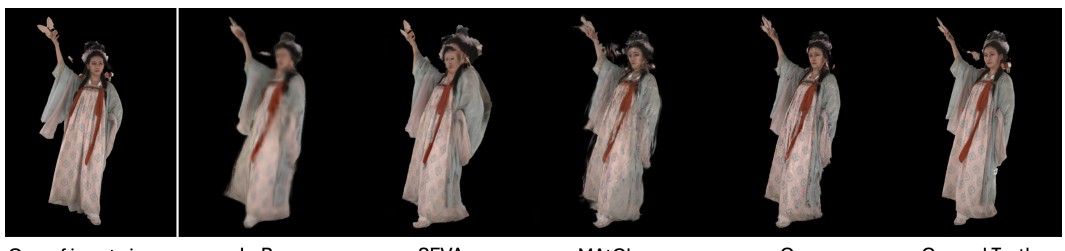

One of input views       LaRa       SEVA       MAtCha       Ours       Ground Truth

Figure 5: **Novel View Synthesis from 4 Views.** We show qualitative results for novel view synthesis on the DNA-Rendering test set. Benefiting from our accurate reconstruction, we achieve photorealistic rendering while avoiding issues present in baselines, including overly smooth appearance (LaRa), hallucinated textures (SEVA), and floater artifacts (MAtCha). Please refer to the appendix for more qualitative results.

reconstructions by leveraging much denser body-attribute predictions. These dense cues help the model disambiguate challenging regions under occlusion or loose clothing, while also capturing fine-grained hands/feet poses, leading to more accurate body estimation. Quantitatively, it consistently outperforms all baselines on both in-domain and cross-domain test sets.

### 4.4 NOVEL VIEW SYNTHESIS

**Baselines.** We compare our method with LaRa (Chen et al., 2024a), SEVA (Zhou et al., 2025b), and MAtCha (Guédon et al., 2025). We report direct inference results with pretrained SEVA due to the lack of training code. We provide an additional comparison with GHG (Kwon et al., 2024) for novel view synthesis in the appendix.

**Metrics.** We evaluate the rendering qualities with four standard metrics: PSNR, SSIM (Wang et al., 2004), LPIPS (Zhang et al., 2018), and FID (Heusel et al., 2017).

**Discussion.** Fig. 5 and Tab. 3 present qualitative and quantitative comparisons for novel view synthesis. LaRa produces overly blurry renderings due to limited volume resolution, while SEVA generates realistic textures but often over-hallucinates. MAtCha achieves photorealistic results but suffers from floating artifacts caused by degenerated charts optimization results; constraining Gaussian positions less reduces this issue but leads to overfitting to training views. In contrast, our method produces sharper details and higher visual fidelity by initializing Gaussians from accurate clothed surfaces. This is also reflected by superior quantitative performance on all metrics.

## 5 CONCLUSION

In this paper, we presented HART, a unified framework for clothed mesh reconstruction, SMPL-X estimation, and novel view synthesis from sparse, uncalibrated human images. It jointly predicts per-pixel point maps, normals, and SMPL-X attributes, enabling recovery of both clothed and body meshes, and facilitating downstream applications such as novel-view synthesis. Extensive experiments demonstrate that HART consistently outperforms state-of-the-art baselines across all tasks.

**Limitations & Future Work:** While effective, our reconstructions still lack fine-scale details (e.g., fingers, hair) due to limited indicator grid resolutions. Rendering qualities also degrade significantly under very sparse views (e.g., 3 views) or challenging lighting. Future work could explore hierarchical or multi-scale architectures for detail recovery, diffusion priors for improved rendering of occluded regions, and video-based training to enhance temporal consistency and enable animatable reconstructions.

## 6 ETHICS STATEMENT

Our work advances human reconstruction, including mesh recovery and novel view synthesis. These contributions hold potential to benefit diverse applications in AR/VR, virtual try-on, and telepresence, fostering progress in both research and real-world use. However, we acknowledge that improving the photorealism and robustness of human reconstruction techniques may also indirectly facilitate misuse, such as the creation of deep fakes or synthetic human content without consent. We emphasize that our models and datasets are intended solely for legitimate academic and industrial research, and we encourage responsible use of the released code and models.

## 7 REPRODUCIBILITY

To promote openness and ensure reproducibility, we provide comprehensive resources for replicating our results: 1) **Open-source code.** We will release the complete source code used in our experiments, including detailed documentation and preprocessing scripts for constructing the training and test sets, along with step-by-step instructions for reproducing the main results. 2) **Pre-trained models.** To facilitate verification and support downstream research, we will publicly release our trained models.

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

# A APPENDIX

## A.1 MORE DETAILS ABOUT OUR SMPL-X HEADS

As detailed in Sec. 3.1, our transformer contains a total of 3 SMPL-X DPT heads: tightness direction head, tightness magnitude head, and body part label head.

### A.1.1 TIGHTNESS DIRECTION AND MAGNITUDE HEADS

For the tightness direction head, we predict a 3D vector field $\hat{\mathbf{d}}_i \in \mathbb{R}^{3 \times H \times W}$, where each vector points from the per-pixel point map $\hat{p}_i$ toward the nearest surface point of the underlying SMPL-X body mesh. The tightness magnitude head predicts a scalar field $\hat{b}_i \in \mathbb{R}^{H \times W}$, representing the lengths of these vectors. Together, they form the full tightness vectors $\hat{\mathbf{v}}_i = \hat{b}_i \hat{\mathbf{d}}_i$, which we use to compute inner body points $\hat{\mathbf{y}}_i = \hat{p}_i + \hat{\mathbf{v}}_i$ for marker aggregation and SMPL-X fitting.

Unlike ETCH (Li et al., 2025), which enforces SE(3) equivariance with a fixed SO(3) anchor array to improve generalization, we directly predict the 3D directions. Since our formulation relies on 2D features from ViT encoders rather than per-point 3D features, enforcing strict equivariance provides limited benefit in our setting.

### A.1.2 BODY PART LABEL HEAD

Another key attribute for marker-based SMPL-X fitting is provided by our body part label head, which assigns each clothed surface point to one of 86 predefined SMPL-X body markers.

The label head predicts two sets of logits using a DPT decoder: 1) an 86-dimensional classification vector $\mathbf{z}_i \in \mathbb{R}^{86}$, and 2) an 86-dimensional confidence vector $\mathbf{c}_i \in \mathbb{R}^{86}$.

The classification logits are normalized via softmax to produce a per-pixel probability distribution:

$$p_{i,k} = \frac{\exp(\mathbf{z}_{i,k})}{\sum_{k'=1}^{86} \exp(\mathbf{z}_{i,k'})}. \tag{11}$$

In parallel, the confidence scores $\mathbf{c}_i$ provide uncertainty estimates. Following (Li et al., 2025; Bhatnagar et al., 2020), we compute the aggregated confidence $\hat{c}_i$ as:

$$\hat{c}_i = \sum_{k=1}^{86} p_{i,k} \cdot \mathbf{c}_{i,k}. \tag{12}$$

The final body part label assignment is obtained as the most probable class:

$$\hat{l}_i = \arg \max_{k \in \{1, \ldots, 86\}} p_{i,k}. \tag{13}$$

Thus, the body part label head produces a feature map of shape $\mathbb{R}^{172 \times H \times W}$, which encodes both classification probabilities and per-label confidences. These are aggregated into per-pixel label and confidence maps, $\hat{l}_i \in \mathbb{N}^{H \times W}$ and $\hat{c}_i \in \mathbb{R}^{H \times W}$, enabling reliable body part assignment and uncertainty-aware aggregation for the subsequent maker-based SMPL-X fitting.

## A.2 ARCHITECTURE OF OUR INDICATOR GRID REFINEMENT MODULE

As discussed in 3.2, we integrate a 3D U-Net into the Differentiable Poisson Surface Reconstruction (DPSR) module to refine the indicator grid and address self-occlusions. The architecture of this refinement module is detailed in Tab. 4. Because our indicator grid $\chi_0$ has a high resolution ($512 \times 512 \times 512$), directly applying a 3D U-Net leads to out-of-memory issues. To overcome this, we first downsample the grid by a factor of 4 using convolutional layers with nonlinear activations. The downsampled grid is then processed with a 3D U-Net, and the output is upsampled back to the original resolution to form the residual grid prediction $\chi_{\text{res}}$. We further observe that using deconvolutional layers in the upsampling module introduces checkerboard artifacts, resulting in noisy surfaces in occluded regions. To avoid this, we adopt convolutional and trilinear interpolation-based upsampling layers, which yield smoother and more accurate surface reconstructions.

Table 4: **Architecture of our Indicator Grid Refinement module.** The module consists of a downsampling block, a 3D U-Net (Çiçek et al., 2016) backbone, and an upsampling block. Starting from the initial indicator grid $\chi_0$, obtained by applying DPSR to the predicted per-pixel oriented point maps $(\mathbf{P}, \mathbf{N})$, we first downsample the grid by a factor of 4. The downsampled grid is processed by the 3D U-Net, and then upsampled back to the original resolution to produce the residual indicator grid $\chi_{\text{res}}$.

| # | Layer Description | Output Dim. |
|---|---|---|
| | **Input** | |
| – | Initial indicator grid $\chi_0$ | $D \times H \times W \times 1$ |
| | **Downsample** | |
| 1 | (4 × 4 × 4 conv, 16 features, stride 2), ReLU | $\frac{1}{2}D \times \frac{1}{2}H \times \frac{1}{2}W \times 16$ |
| 2 | (4 × 4 × 4 conv, 32 features, stride 2), ReLU | $\frac{1}{4}D \times \frac{1}{4}H \times \frac{1}{4}W \times 32$ |
| | **3D U-Net** | |
| 3 | Encoder: (3 × 3 × 3 conv, 32 features, stride 1) × 2 | $\frac{1}{4}D \times \frac{1}{4}H \times \frac{1}{4}W \times 32$ |
| 4 | Encoder: (3 × 3 × 3 conv, 64 features, stride 2) | $\frac{1}{8}D \times \frac{1}{8}H \times \frac{1}{8}W \times 64$ |
| 5 | Encoder: (3 × 3 × 3 conv, 128 features, stride 2) | $\frac{1}{16}D \times \frac{1}{16}H \times \frac{1}{16}W \times 128$ |
| 6 | Decoder: Upsample ×2 + (3 × 3 × 3 conv, 64 features, stride 1) × 2 | $\frac{1}{8}D \times \frac{1}{8}H \times \frac{1}{8}W \times 64$ |
| 7 | Decoder: Upsample ×2 + (3 × 3 × 3 conv, 32 features, stride 1) × 2 | $\frac{1}{4}D \times \frac{1}{4}H \times \frac{1}{4}W \times 32$ |
| 8 | Final (1 × 1 × 1 conv, 32 features, stride 1) | $\frac{1}{4}D \times \frac{1}{4}H \times \frac{1}{4}W \times 32$ |
| | **Upsample** | |
| 9 | (3 × 3 × 3 conv, 16 features, stride 1), ReLU | $\frac{1}{4}D \times \frac{1}{4}H \times \frac{1}{4}W \times 16$ |
| 10 | Trilinear Upsample ×2 | $\frac{1}{2}D \times \frac{1}{2}H \times \frac{1}{2}W \times 16$ |
| 11 | (3 × 3 × 3 conv, 8 features, stride 1), ReLU | $\frac{1}{2}D \times \frac{1}{2}H \times \frac{1}{2}W \times 8$ |
| 12 | Trilinear Upsample ×2 | $D \times H \times W \times 8$ |
| 13 | (3 × 3 × 3 conv, 1 feature, stride 1) | $D \times H \times W \times 1$ |
| | **Output** | |
| – | Residual indicator grid $\chi_{\text{res}}$ | $D \times H \times W \times 1$ |

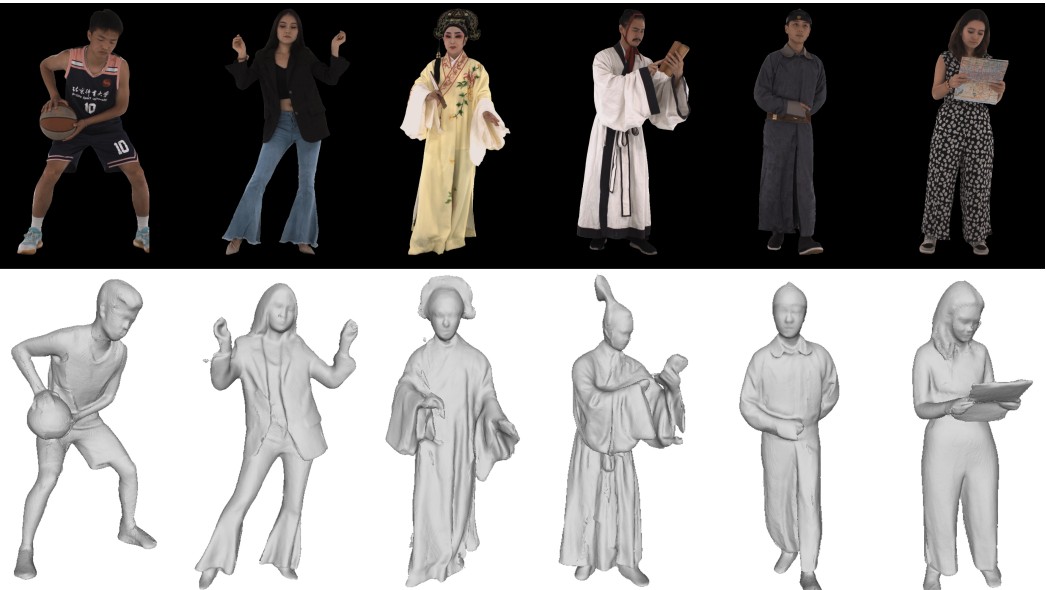

Figure 6: **Qualitative Results on Clothed Mesh Reconstruction from the DNA-Rendering Test Set.** We show one of the 4 input images in row 1 and our reconstructed meshes in row 2. Although trained only on synthetic human scans, our method generalizes effectively to real-world images, producing accurate clothed meshes even under challenging conditions with complex garments and human–object interactions.

Table 5: **Effect of the number of input views.** Performance consistently improves as the number of input views increases across all three tasks. With only 3 views, the reconstructions and renderings already achieve decent scores, but increasing to 4 or more views yields notable gains in geometry completeness, SMPL-X robustness, and novel view fidelity. The best results are obtained with 8 input views, where reconstructions are most complete and renderings most photorealistic.

| Number of input views | Clothed Mesh Reconstruction | | | | | SMPL-X Estimation | | Novel View Synthesis | | | |
|---|---|---|---|---|---|---|---|---|---|---|---|
| | Acc. | Comp. | CD | F-Score | NC | PA-V2V | PA-MPJPE | PSNR | SSIM | LPIPS | FID |
| 3 views | 0.0088 | 0.0130 | 0.0218 | 0.9041 | 0.9057 | 17.66 | 18.13 | 30.46 | 0.9585 | 0.0481 | 22.02 |
| 4 views | 0.0067 | 0.0105 | 0.0172 | 0.9354 | 0.9125 | 16.96 | 17.67 | 31.70 | 0.9675 | 0.0390 | 14.24 |
| 6 views | 0.0049 | 0.0083 | 0.0132 | 0.9611 | 0.9200 | 16.87 | 17.53 | 34.06 | 0.9799 | 0.0244 | 5.42 |
| 8 views | **0.0044** | **0.0077** | **0.0121** | **0.9675** | **0.9229** | **16.56** | **17.31** | **35.11** | **0.9833** | **0.0199** | **4.04** |

## A.3 QUALITATIVE RESULTS FOR CLOTHED MESH RECONSTRUCTION ON DNA-RENDERING

As shown in Fig. 6, our method successfully reconstructs clothed meshes with accurate geometry even in challenging scenarios involving complex garments and human–object interactions, highlighting its robustness across domains.

## A.4 ADDITIONAL DETAILS FOR BASELINE SETUPS

We provide additional details for baseline setups for our 3 downstream tasks.

**Clothed Mesh Reconstruction.** For our method and VGGT (Wang et al., 2025), the predicted geometries are aligned with the ground truth via the Umeyama (Umeyama, 1991) algorithm at the point map level. To ensure fairness under uncalibrated settings, we use the camera parameters estimated by our method rather than ground-truth for LaRa (Chen et al., 2024a) and MAtCha (Guédon et al., 2025), and apply the same Umeyama solution to align their predicted meshes with the ground truth. For Puzzle Avatar (Xiu et al., 2024), since the human mesh is optimized in SMPL-X A-pose, we use ground-truth SMPL-X parameters to perform nearest-neighbor SMPL skinning to warp the canonical mesh into posed space, which also roughly aligns the warped clothed mesh with the ground-truth clothed mesh.

**SMPL-X Estimation.** Both EasyMocap (eas, 2021; Shuai et al., 2022) and MV-SMPLify-X (Pavlakos et al., 2019; Zheng et al., 2021) rely on keypoint fitting. For fair comparisons under uncalibrated settings, we also use the camera parameters estimated by our method for keypoint triangulation and projection.

Note that ETCH (Li et al., 2025) originally does not use shape or pose regularizations during marker fitting. For fair comparison, we also use the same regularizations as in our method.

As with our method, we assume that the global scale of the scene is unknown. Consequently, we also optimize the SMPL-X scale for the baselines. For EasyMocap and Multi-view SMPLify-X, the scale is jointly optimized with the other SMPL-X parameters. In contrast, for ETCH, we observed that optimizing the scale in the same manner fails to converge, likely due to the sparsity of the sampled points it can process. To address this, we first normalize the height of all input meshes to 1.7 m and then optimize the remaining SMPL-X parameters.

**Novel View Synthesis.** We construct the DNA-Rendering (Cheng et al., 2023) test set using one frame from each of the 47 subjects in parts 0 and 1, excluding 6 subjects interacting with thin-structured objects, thus having unreliable foreground masks. For each subject, we use the 16 horizontal views and use either 4/6/8 views as inputs to our model, while the rest is held out for evaluation. Following (Jin et al., 2025), we re-estimate color correction matrices and obtain improved segmentation masks by voting with multiple segmentation models (Lin et al., 2021; Zheng et al., 2024).

To align our predicted clothed meshes with the ground-truth cameras for the evaluation purpose, we train 2DGS (Huang et al., 2024) on all 16 views and render depths from all these views from the optimized Gaussians. These depth maps serve as pseudo ground-truth for aligning our predicted geometry via Umeyama alignment.

Table 6: **Quantitative results on 6-view and 8-view Novel View Synthesis.** Under higher number of input views, our method still consistently outperform MAtCha on both synthetic and real-world test sets.

| Methods | THuman 2.1 (Synthetic) | | | | DNA Rendering (Real World) | | | |
|---|---|---|---|---|---|---|---|---|
| | PSNR | SSIM | LPIPS | FID | PSNR | SSIM | LPIPS | FID |
| **6 input views** | | | | | | | | |
| MAtCha (Guédon et al., 2025) | 33.08 | 0.9750 | 0.0317 | 7.05 | 27.44 | 0.9332 | 0.0593 | **29.71** |
| Ours | **34.06** | **0.9799** | **0.0244** | **5.42** | **28.44** | **0.9449** | **0.0522** | 30.42 |
| **8 input views** | | | | | | | | |
| MAtCha (Guédon et al., 2025) | 34.34 | 0.9810 | 0.0237 | 4.50 | 27.75 | 0.9389 | 0.0520 | 25.04 |
| Ours | **35.11** | **0.9833** | **0.0199** | **4.04** | **28.86** | **0.9502** | **0.0455** | **24.56** |

Table 7: **Quantitative results on 3-view Novel View Synthesis.** Compared to GHG (Kwon et al., 2024), our method achieves higher PSNR, SSIM, and lower LPIPS, indicating more faithful and perceptually realistic renderings, while GHG attains a slightly lower FID due to its use of a diffusion-based inpainting model.

| Methods | THuman 2.1 (Synthetic) | | | |
|---|---|---|---|---|
| | PSNR | SSIM | LPIPS | FID |
| GHG (Kwon et al., 2024) | 23.46 | 0.9181 | 0.0740 | **19.04** |
| Ours | **27.10** | **0.9480** | **0.0574** | 22.19 |

## A.5 EFFECT OF NUMBER OF INPUT VIEWS

Tab. 5 shows the effect of the number of input views on our method across all 3 downstream tasks. We observe consistent performance gains as the number of input views increases. Even with only 3 views, our method already produces competitive quantitative scores, and adding more views steadily improves the metrics across all tasks. Moving from 3 to 4 views yields clear improvements across all metrics, showing the benefit of increasing viewpoint coverage. With 6 and 8 views, reconstruction errors drop further, SMPL-X estimation becomes more accurate, and novel view synthesis achieves higher fidelity with fewer artifacts. Using 8 views achieves the best overall performance, as denser inputs help resolve occlusions and capture finer details.

For qualitative results on novel view synthesis with different numbers of input views, please refer to Sec. A.6.

## A.6 ADDITIONAL RESULTS FOR NOVEL VIEW SYNTHESIS (NVS)

### A.6.1 ADDITIONAL COMPARISONS ON DIFFERENT NUMBER OF INPUT VIEWS

As shown in the top part of Fig. 7, we provide additional 4-view NVS results comparing with LaRa (Chen et al., 2024a), SEVA (Zhou et al., 2025b), and MAtCha (Guédon et al., 2025). To further demonstrate robustness under varying numbers of input views, we compare against MAtCha—the most competitive baseline for NVS—under 6- and 8-view settings. The bottom part of Fig. 7 and Tab. 6 show that our method achieves consistent improvements across most metrics on both synthetic and real-world test sets. While MAtCha's results improve with more views, its quality remains limited by inaccurate geometry initialization from its aligned charts, and its strategy of allowing Gaussians to move more freely (with densification and pruning) [1] makes it more prone to overfitting training views. In contrast, our method leverages accurate clothed-mesh constraints, producing higher-quality and more faithful novel view renderings.

### A.6.2 COMPARISON WITH GHG ON 3 INPUT VIEWS

We provide an additional comparison with GHG (Kwon et al., 2024) for novel view synthesis under the 3-view setting. GHG tends to overfit to the training camera distributions and produces misaligned renderings on real-world inputs. For this reason, we restrict the comparison to the THuman test set, using its official test set. Since GHG was trained on only a subset of THuman 2.1 subjects, we train our model on the same reduced set for fairness and evaluate against the released GHG pretrained

---

[1]Although MAtCha paper claims Gaussian positions and covariances are fixed, their official code still optimizes them, and we find that learning all gaussian attributes and enabling gaussian densification/pruning consistently improves their performance.

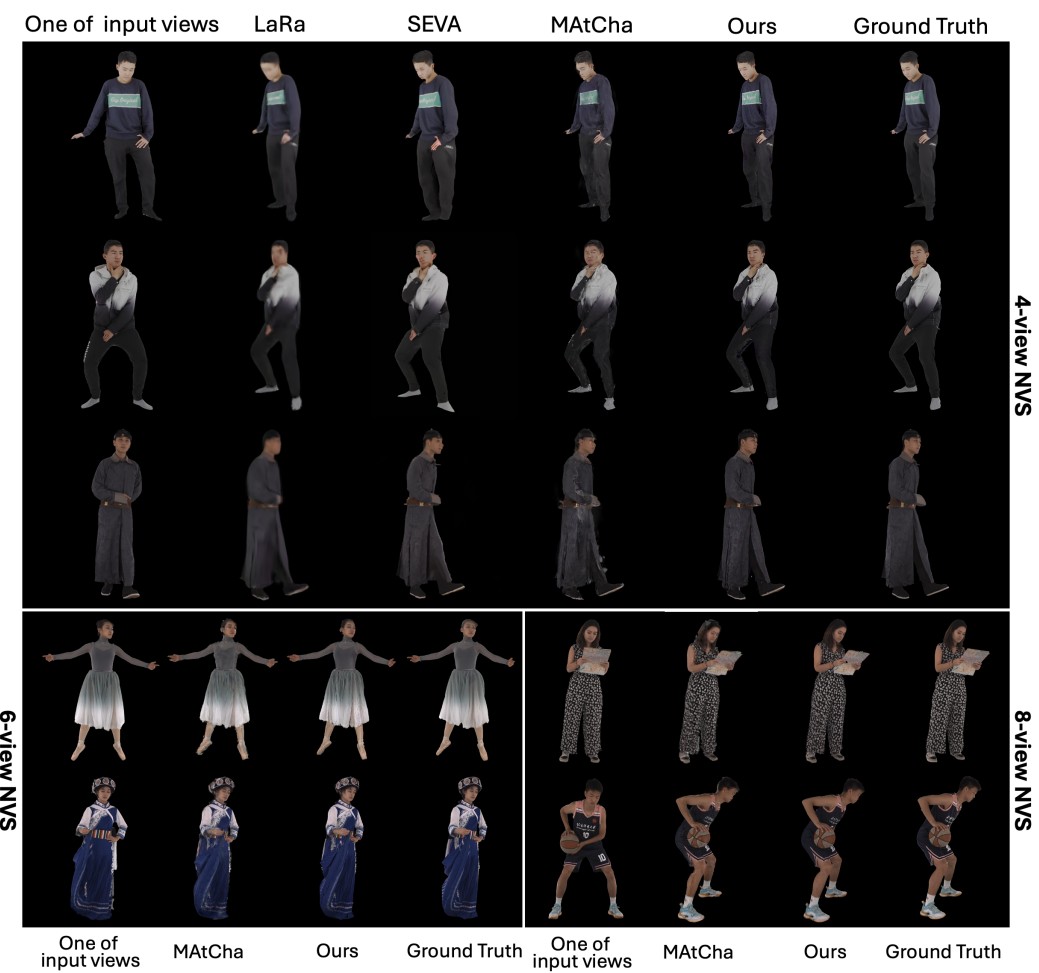

Figure 7: **Additional Qualitative Results on 4-view, 6-view and 8-view Novel View Synthesis.** We present results on two THuman subjects and one DNA-Rendering subject for 4-view NVS, comparing against LaRa (Chen et al., 2024a), SEVA(Zhou et al., 2025b), and MAtCha(Guédon et al., 2025). As discussed in Fig. 5, our method consistently produces higher-quality renderings than all baselines. We further compare with MAtCha under 6-view and 8-view settings on DNA-Rendering subjects. While MAtCha's results improve with more input views and achieve photorealistic renderings, it continues to suffer from floating artifacts due to less reliable Gaussian initialization from charts. In contrast, our method delivers renderings that more closely align with the ground truth.

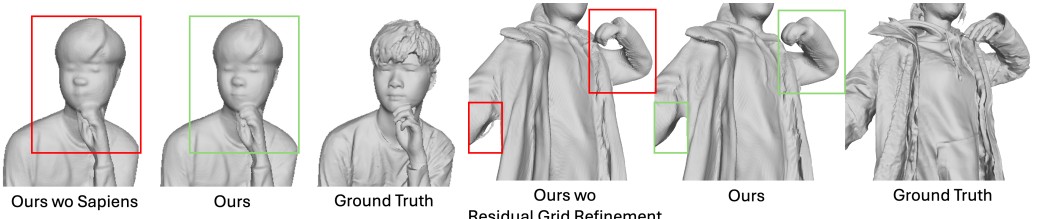

| One of input views | GHG | Ours | Ground Truth |

Figure 8: **Qualitative Results on 3-view Novel View Synthesis.** Although GHG produces photo-realistic novel views, it fails to recover the correct body shape in loose garments due to its reliance on SMPL mesh, and it occasionally produces incorrect poses due to errors in SMPL estimation. Our method, on the other hand, recovers the body shape better due to the initialization from more accurate clothed geometry.

| Ours wo Sapiens | Ours | Ground Truth | Ours wo Residual Grid Refinement | Ours | Ground Truth |

Figure 9: **Ablation Studies.** Removing Sapiens (Khirodkar et al., 2024) normals results in blurrier surfaces, while removing the indicator grid refinement leads to incomplete and less accurate geometry in self-occluded regions. Our full method produces the most detailed, accurate, and complete meshes.

model. As GHG performs best when using ground-truth camera parameters, we adopt the same setup for our method during the novel view synthesis part in this comparison.

As shown in Fig. 8 and Tab. 7, GHG relies on a SMPL body mesh as the template, which struggles to recover accurate body shapes under loose garments, and leads to inaccurate body poses due to errors in SMPL estimations. In contrast, our method renders more faithful body shapes by leveraging a more accurate clothed-mesh initialization.

## A.7 ABLATION STUDIES

We ablate two critical design choices on the 100 test subjects of the THuman 2.1 dataset. First, we remove the Sapiens model (Khirodkar et al., 2024) for base normal prediction and instead train the network to regress full normal maps from scratch; this variant is denoted as *Ours w/o $\mathcal{S}$*. Second, we disable indicator grid refinement and reconstruct the mesh directly from the initial indicator grid $\chi_0$ obtained from per-pixel oriented point maps; this variant is denoted as *Ours w/o $\mathcal{R}$*. Finally, we disable both Sapiens normals and indicator-grid refinement (denoted *Ours w/o $\mathcal{S}$, w/o $\mathcal{R}$*), which is essentially equivalent to fine-tuning VGGT (Wang et al., 2025) with a DPSR head. For each subject,

Table 8: **Ablation Studies on Clothed Geometry.** Removing the Sapiens normals ($\mathcal{S}$) and/or the indicator grid refinement ($\mathcal{R}$) degrades reconstruction accuracy, highlighting their importance.

| Methods | $\mathcal{S}$ | $\mathcal{R}$ | Acc. ↓ | Comp. ↓ | CD ↓ | F-Score ↑ | NC ↑ |
|---|---|---|---|---|---|---|---|
| Ours w/o $\mathcal{S}$ | ☒ | ☐ | 0.0073±0.0003 | 0.0114±0.0008 | 0.0187±0.0010 | 0.9272±0.0032 | 0.9071±0.0023 |
| Ours w/o $\mathcal{R}$ | ☐ | ☒ | 0.0087±0.0006 | 0.0138±0.0010 | 0.0224±0.0014 | 0.9156±0.0035 | 0.9022±0.0024 |
| Ours w/o $\mathcal{S}$, w/o $\mathcal{R}$ | ☒ | ☒ | 0.0091±0.0005 | 0.0144±0.0010 | 0.0235±0.0015 | 0.9083±0.0035 | 0.8970±0.0025 |
| Ours | ☐ | ☐ | **0.0067**±0.0003 | **0.0104**±0.0007 | **0.0171**±0.0009 | **0.9360**±0.0032 | **0.9125**±0.0022 |

Table 9: **Quantitative Evaluations on Per-pixel Normal Maps.** We report cosine similarity and L1 loss.

| Methods | Cosine Similarity ↑ | L1 Loss ↓ |
|---|---|---|
| Ours w/o Sapiens | 0.9548 | 0.1131 |
| Sapiens Monocular Normals (Khirodkar et al., 2024) | 0.9553 | 0.1148 |
| Ours | **0.9671** | **0.0951** |

we randomly sample 4 input views, and for every entry we report the mean ± standard error of the mean (SEM) to additionally quantify the uncertainty of the estimated mean.

As shown in Tab. 8 and Fig. 9, predicting normals entirely from scratch, rather than as residuals to Sapiens normals, results in blurrier surfaces and a clear drop across all metrics. Similarly, removing the residual grid refinement reduces the pipeline to standard Poisson-based reconstruction on VGGT point maps, which leads to even more significant performance degradation and inaccurate geometry in occluded regions. Our full model consistently outperforms all ablated variants across all metrics. For higher-the-better metrics (F-Score, NC), the lower bound of our full model (mean - SEM) is already above the upper bound of each ablated variant (mean + SEM), while for lower-the-better metrics (Acc., Comp., CD), the upper bound of our model (mean + SEM) is mostly below the lower bound of the ablations (mean - SEM). This separation of mean ± SEM intervals indicates that the improvements from the Sapiens prior and indicator grid refinement are consistent and unlikely to be due to sampling noise.

### A.8 IMPLEMENTATION DETAILS

Our network operates at an input image resolution of $518 \times 518$ and an indicator grid resolution of $512 \times 512 \times 512$. We finetune the model from the pretrained VGGT-1B checkpoint, using an initial learning rate of 1e-4 for the SMPL-X branch (trained from scratch) and 1e-5 for the remaining modules, with cosine decay down to a minimum of 1e-6. To stabilize training, we apply gradient norm clipping at 0.5. For efficiency, we use bfloat16 precision and gradient checkpointing to reduce GPU memory usage. Unlike VGGT, we additionally freeze the DINO encoder to further save memory. To improve the generalization on different numbers of input views, we randomly alternate between 3 to 8 views during training. We train the network for 20 epochs with 10,000 steps per epoch, which takes approximately 50 hours with 8 NVIDIA L40S GPUs.

Our SMPL-X marker fitting uses a Gauss–Newton optimizer similar to (Li et al., 2025), implemented with the Levenberg–Marquardt algorithm (Roweis, 1996). The optimization is performed in two stages: in the first stage, we optimize the poses along with the first two shape coefficients, and in the second stage, we additionally optimize the remaining shape coefficients. Thanks to the lightweight marker formulation, the entire fitting procedure converges within only a few seconds.

Following (Guédon et al., 2025), our geometry-informed novel view synthesis optimizes the 2D gaussians for 7,000 steps, which takes approximately 5 minutes on a single L40S GPU.

### A.9 LLM USAGE DISCLOSURE

In this manuscript, we use LLMs for grammar polishing and sentence-level structure refinement.

### A.10 EVALUATION ON NORMAL MAPS

We provide both quantitative and qualitative evaluations of the predicted normal maps. We compare three variants: (i) *Ours w/o Sapiens*, which predicts normals from scratch without using the Sapiens prior; (ii) *Sapiens Monocular Normals*, which directly evaluates Sapiens monocular normals predictions without applying our refinement head; and (iii) *Ours*, which predicts residuals on top of Sapiens normals and refines them in a multi-view–consistent manner.

Fig. 10 visualizes the predicted normal maps. *Ours w/o Sapiens* produces over-smoothed normals that lack fine details when trained from scratch. *Sapiens Monocular Normals* capture detailed structures (e.g., facial features) but are not consistent across views. Our full method predicts residual

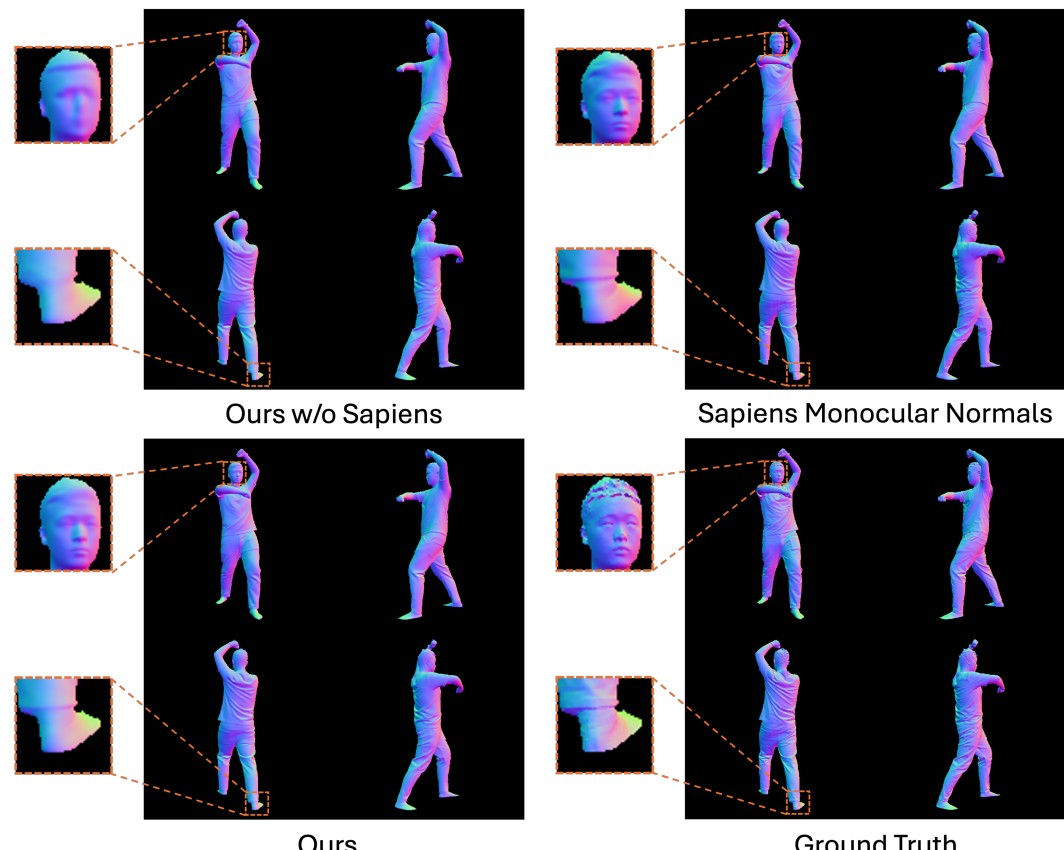

Ours w/o Sapiens      Sapiens Monocular Normals

Ours         Ground Truth

Figure 10: **Qualitative comparison of normal predictions.** Ours w/o Sapiens (top-left), which predicts normals from scratch, produces over-smoothed normals that lack fine details. Sapiens monocular normals (Khirodkar et al., 2024) (top-right) capture detailed facial and clothing structure but are not consistent across views. Our full method (bottom-left) predicts residual normals on top of Sapiens, retaining its details while significantly improving multi-view consistency and align closer with the ground truth normals (bottom-right).

normals on top of the Sapiens prior, retaining its fine details while significantly improving multi-view consistency. Tab. 9 further reports the average cosine similarity and L1 error between predicted normals and ground-truth normals on the THuman 2.1 test set. Our full model achieves the best performance on both metrics, indicating that the proposed refinement improves not only visual quality but also quantitative accuracy.

## A.11 COMPARISON WITH SINGLE-VIEW METHODS

To prove that our multi-view reconstruction approach outperforms single-view methods by benefiting from more views, we perform an addition comparison to the state-of-the-art single-view clothed human reconstruction method PSHuman (Li et al., 2024a). We directly evaluate PSHuman using its released checkpoint, as it has already been trained on the same THuman 2.1 dataset. For each test subject, we use a single front view as input to PSHuman and align its prediction to the ground-truth mesh using ICP before evaluation. Since our method is not designed for single-view hallucination (we do not adopt a diffusion prior to infer unseen geometry), we report results with 2, 3, and 4 views, always including the front view. Although our model is only trained with 3-8 views inputs, it still generalizes to 2-view settings.

As shown in Fig. 11 and Tab. 10, our method significantly outperforms PSHuman with as few as 2 views in terms of reconstruction fidelity and alignment with ground truth. In practice, we observe that although PSHuman can hallucinate fine details with a high-resolution multi-view dif-

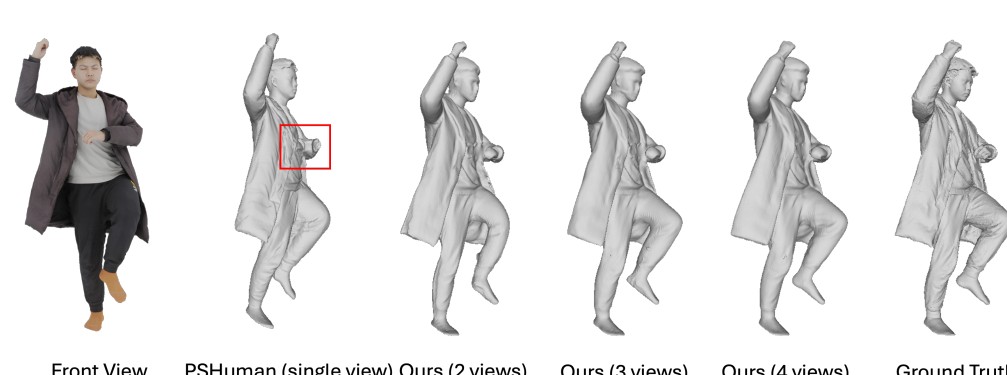

Front View    PSHuman (single view) Ours (2 views)    Ours (3 views)    Ours (4 views)    Ground Truth

Figure 11: **Comparison with the single-view baseline PSHuman (Li et al., 2024a) and our method under different numbers of input views.** For results, we include the front view as one of the input images. PSHuman can hallucinate fine local details such as faces but often fails to faithfully recover the true body pose and shape from a single image, leading to noticeable misalignment with the ground truth and inaccurate body parts (e.g., arm region in red box). In contrast, our multi-view approach benefits from additional views to faithfully reconstruct the actual geometry rather than hallucinating unseen regions: with as few as 2 views (front + back) it already reconstructs the mesh that aligns well with the ground truth.

Table 10: **Quantitative comparison between PSHuman (Li et al., 2024a) and our method under different numbers of input views.** Our method benefits from multi-view inputs to faithfully reconstruct the clothed human geometry and significantly outperform the PSHuman reconstructions from single views.

| Method | Acc. ↓ | Comp. ↓ | CD ↓ | F-Score ↑ | NC ↑ |
|---|---|---|---|---|---|
| PSHuman (1 view) | 0.1862 | 0.4103 | 0.5965 | 0.3649 | 0.7407 |
| Ours (2 views) | 0.0420 | 0.0630 | 0.1050 | 0.6420 | 0.8609 |
| Ours (3 views) | 0.0088 | 0.0130 | 0.0218 | 0.9041 | 0.9057 |
| Ours (4 views) | 0.0067 | 0.0105 | 0.0172 | 0.9354 | 0.9125 |

fusion model, it often fails to faithfully reconstruct the true body pose and shape from a single image, leading to noticeable misalignment with the ground-truth mesh even after ICP alignment. In contrast, our method leverages multiple views to faithfully recover the actual geometry rather than hallucinating unseen regions. As expected, our performance improves as more views are provided: 2-view reconstructions (front + back) are already substantially better than PSHuman, and 3/4-view reconstructions further improves the quantitative scores.

### A.12 EVALUATION ON CAMERA ESTIMATIONS

Although we adopt VGGT as our backbone, we intentionally avoid using its feed-forward camera head because it is pretrained assuming centered principal points, which holds for many scene-level datasets but not for human datasets where the foreground subject is often off-center. In our setting, we need to center the human bounding box to maximize foreground coverage and training stability; this changes the effective principal point, making the original VGGT camera head not suitable for our data. Instead, we follow a standard camera estimation pipeline introduced in Dust3R (Wang et al., 2024b) and adopted by follow-up works such as Fast3R (Yang et al., 2025): focal lengths are recovered with the Weiszfeld algorithm (Plastria, 2011), and extrinsics are recovered by PnP-RANSAC (Lepetit et al., 2009; Fischler & Bolles, 1981) applied to dense pointmaps.

To systematically compare our camera estimations against VGGT camera head predictions, we re-render THuman 2.1 so that both the principal point and the human bounding box are centered:

Table 11: **Camera accuracy comparison between VGGT camera head (Wang et al., 2025) and our PnP-RANSAC-based cameras.** We evaluate R/T pose accuracy (AUC@30) and 2D Reprojection Accuracy (AUC@10px) on both in-domain centered principal points test set and the cross-domain centered human bounding boxes test set. Our method achieves camera accuracy on par with VGGT on in-domain test, while producing significantly better aligned 2D projections on the cross-domain test set.

| Methods | Centered principal points | | Centered human bbox |
| --- | --- | --- | --- |
| | R/T AUC@30 ↑ | 2D Reproj. AUC@10px ↑ | 2D Reproj. AUC@10px ↑ |
| VGGT camera head | **99.97** | 97.73 | 12.13 |
| Ours | 96.29 | **99.49** | **95.28** |

we keep the principal point fixed at the image center, and shift the camera translations in 3D to compensate for centering the human bounding box in 2D. We retrain both methods on this controlled dataset. We then evaluate on two test splits with the same 100 subjects: (i) Test set 1 (centered principal points): synthetic images where the GT principal points are centered, and (ii) Test set 2 (off-center principal points / centered human bounding boxes): our original THuman 2.1 test set, where we center the human bounding box at inference (more aligned with real-world inputs). We report (i) R/T AUC@30 on test set 1, following VGGT, to mesure the extrinsics accuracy, and (ii) 2D reprojection AUC@10px, which measures how well the 2D projections of predicted pointmaps align with the GT foreground pixels, which is critical for the subsequent novel view synthesis task. We only report R/T AUC@30 on test set 1, where the ground-truth principal points are centered and this metric is well-defined; on test set 2, the ground-truth intrinsics are off-centered while our estimated intrinsics are constrained to have centered principal points, so pose error alone is not diagnostic of camera accuracy.

As shown in Tab. 11, our PnP-based cameras are on par with VGGT on the ideal and in-distribution centered setting, while providing dramatically better 2D alignment when evaluated under out-of-distribution and off-center settings that are closer to real world settings. In contrast, the VGGT camera head overfits to the training distribution with centered principal points and produces significantly misaligned 2D projections when tested on off-center settings. Since accurate 2D reprojection is what most directly affects novel view synthesis, our choice of the PnP-RANSAC camera estimation actually reduces error accumulation in the downstream tasks.

### A.13 TRAINING WITH REAL-WORLD DATA

We conduct an additional experiment to evaluate how HART behaves when trained with real-world data, following the mixed synthetic–real training strategy also used in VGGT (Wang et al., 2025). To the best of our knowledge, 4D-Dress (Wang et al., 2024c) is the only real-world multi-view human dataset that provides both accurate clothed meshes and SMPL(-X) annotations, making it suitable for both training and quantitative evaluation, although it contains far fewer distinct subjects than synthetic datasets (32 subjects, each with a tight and a loose outfit). We therefore mix synthetic THuman 2.1 and real 4D-Dress during training: 2 subjects (4 outfits) from 4D-Dress are held out as a real-world test set, and the remaining 30 subjects / 60 outfits are used for training. We sample synthetic and real data with a 10:1 ratio to avoid overfitting to the limited number of real subjects, and train for 20 epochs (11k steps per epoch). At test time, we (i) randomly sample 100 frames from the held-out 4D-Dress subjects (with a frame stride of at least 5) and (ii) evaluate cross-domain generalization on the 2K2K synthetic test set (Han et al., 2023), which is never used for training.

As shown in Tab. 12, Tab. 13, and Fig. 12, we find that incorporating 4D-Dress into the training set significantly improves both clothed-mesh and SMPL-X reconstruction metrics on the 4D-Dress test set. This gain mainly comes from better handling of real-world effects such as strong boundary lighting, which are not well represented in purely synthetic THuman 2.1 renders. In addition, adding real-world data slightly improves cross-domain performance on 2K2K (which is not used for training), suggesting that exposure to loose, diverse garments and complex clothing in 4D-Dress helps our model generalize better to the challenging outfits in 2K2K. We also emphasize that even *without* any training on 4D-Dress, our original model already demonstrates strong generalization to loose

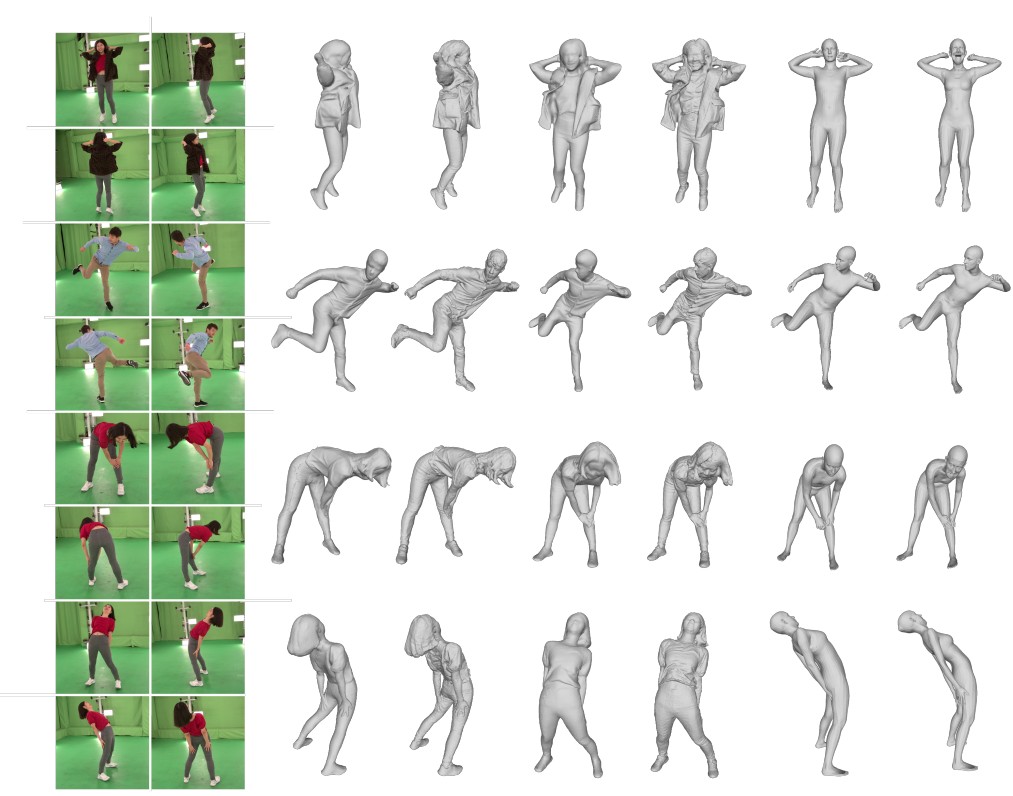

Input Views     Ours View 1   GT View 1   Ours View 2   GT View 2    Our SMPL-X   GT SMPL-X

Figure 12: **Qualitative Results for Clothed Mesh Reconstruction and SMPL-X Estimation on 4D-Dress (Wang et al., 2024c) Test Subjects.** For each frame, we show two viewpoints with ground-truth clothed meshes and one viewpoint with the corresponding ground-truth SMPL-X mesh.

Table 12: **Quantitative Comparison of Models Trained with and without Real-world Data for Clothed Mesh Reconstruction.** Compared to the model trained only on synthetic data (THuman 2.1 (Yu et al., 2021)), the model trained on a mixture of synthetic and real-world data (THuman 2.1 + 4D-Dress (Wang et al., 2024c)) significantly improves reconstruction quality on the 4D-Dress test set, and also slightly improves cross-domain generalization on the 2K2K (Han et al., 2023) test set, which is not used for training.

| Training Dataset(s) | 4D-Dress | | | | | 2K2K | | | | |
|---|---|---|---|---|---|---|---|---|---|---|
| | Acc. | Comp. | CD | F-Score | NC | Acc. | Comp. | CD | F-Score | NC |
| THuman 2.1 (Yu et al., 2021) | 0.0208 | 0.0274 | 0.0482 | 0.7504 | 0.8822 | 0.0077 | 0.0093 | 0.0170 | 0.9301 | 0.9479 |
| THuman 2.1 + 4D-Dress (Wang et al., 2024c) | **0.0085** | **0.0125** | **0.0210** | **0.9029** | **0.9186** | **0.0072** | **0.0089** | **0.0161** | **0.9348** | **0.9498** |

Table 13: **Quantitative Comparison of Models Trained with and without Real-world Data for SMPL-X Estimation.** Consistent with the clothed mesh reconstruction results in Tab. 12, fine-tuning with 4D-Dress (Wang et al., 2024c) significantly improves SMPL-X estimation accuracy on the 4D-Dress test set, and also slightly improves cross-domain generalization on the 2K2K (Han et al., 2023) test set.

| Training Dataset(s) | 4D-Dress | | 2K2K | |
|---|---|---|---|---|
| | PA-V2V | PA-MPJPE | PA-V2V | PA-MPJPE |
| THuman 2.1 (Yu et al., 2021) (Li et al., 2025) | 21.60 | 22.93 | 22.89 | 24.49 |
| THuman 2.1 + 4D-Dress (Wang et al., 2024c) | **19.09** | **18.07** | **22.20** | **23.10** |

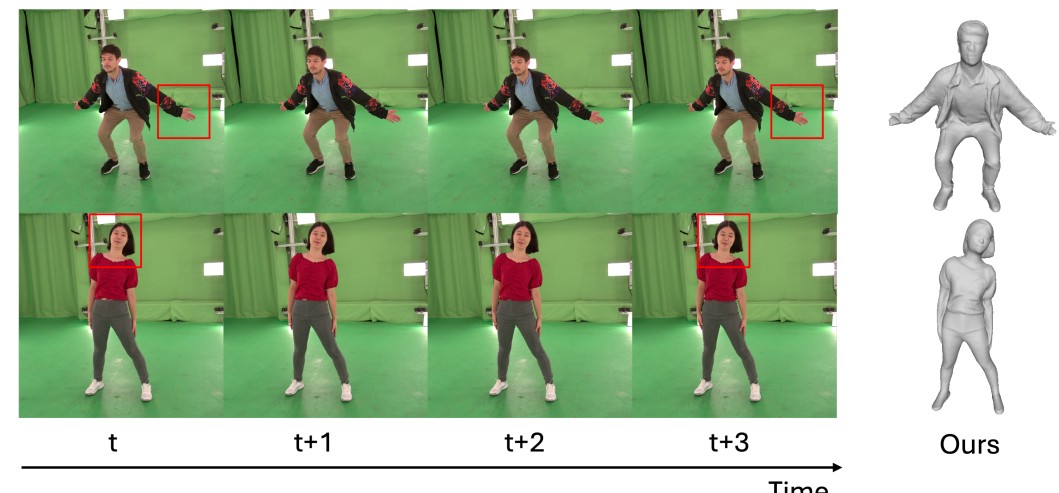

Figure 13: **Robustness to pose misalignment across views.** For each subject, we feed HART with 4 views where each view comes from a *different*, *nearby* timestep of the sequence, introducing small pose and appearance changes between cameras. The reconstructed clothed meshes on the right remain consistent and with only slight artifacts, demonstrating that our method is robust to modest pose misalignments rather than requiring perfectly synchronized captures. We highlight the jittering body parts with red bounding boxes.

clothing and human–object interactions on challenging datasets such as 2K2K and DNA-Rendering (see Fig. 3 and Fig. 6).

### A.14    ROBUSTNESS TO POSE MISALIGNMENTS

In real applications, multi-view images are often acquired sequentially rather than perfectly synchronized, leading to slight pose and appearance changes across views. To test HART under this setting, we construct inputs where each of the four views comes from a *different* but *nearby* timestep of the sequence (i.e., view 1 adopts timestep $t$ of camera 1, view 2 adopts timestep $t + 1$ of camera 2, etc.), introducing small temporal misalignments between cameras. As illustrated in Fig. 13, HART still produces coherent, watertight clothed meshes without significant performance degradation, indicating that our method is robust to modest inter-view pose jitter and does not require strictly synchronized captures.

### A.15    IN-THE-WILD GENERALIZATION

We evaluate our method on a casual in-the-wild capture, where a static subject is recorded using a handheld phone while the user walks around them. As shown in Fig. 14, our model robustly reconstructs a high-fidelity clothed mesh along with the aligned SMPL-X body mesh in a single feed-forward pass.

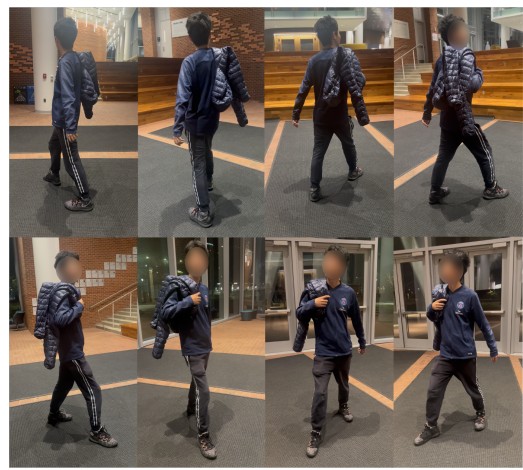 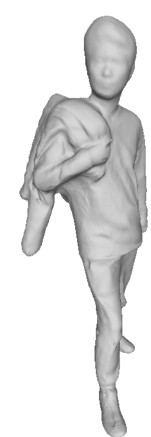 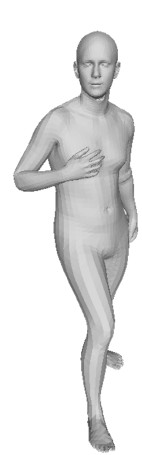

In-the-wild captures          Our clothed mesh          Our SMPL-X

Figure 14: **In-the-wild Generalization.** We demonstrate a 8-view in-the-wild reconstruction on a casual handheld phone capture of a static person: from this input, our model faithfully reconstructs both the clothed mesh and the underlying SMPL-X body mesh in a single feed-forward pass. The face of the subject is blurred in the input views for privacy preservation.

