# OpenReview forum: "HART: Human Aligned Reconstruction Transformer"
_ICLR.cc/2026/Conference — Submitted to ICLR 2026_

### Official Review · Reviewer_iAZE · 2025-10-29

**Soundness:** 3
**Presentation:** 3
**Contribution:** 2
**Rating:** 4
**Confidence:** 4

**Summary:**

HART represents a strong empirical contribution through a well-engineered combination of recent achievements on geometric and human reconstruction. It’s elegant and practical to integrate point-map prediction, occlusion-aware geometry completion, and SMPL-X alignment under a transformer framework.

**Strengths:**

1. The integration of VGGT-based multi-view fusion with per-pixel SMPL-X attribute prediction (tightness vectors and body-part labels) is an interesting adaptation of general 3D reconstruction transformers to the human domain.
2. The paper is clearly structured, with explicit loss formulations, architecture diagrams, and references to code/model release. These increase its reproducibility and transparency.

**Weaknesses:**

1. Although cross-domain generalization is demonstrated, the model is trained entirely on synthetic THuman 2.1 data. How does the model perform when trained or fine-tuned on real-world multi-view captures? Are there domain adaptation or photometric normalization strategies considered?
2. The method heavily depends on VGGT and DPSR. Beyond combining existing modules, what new representational capability does HART itself introduce? Could the same gains be achieved by tuning VGGT + DPSR with human-specific priors?
3. The paper does not provide a detailed sensitivity study on the loss weights or robustness to view number reduction (e.g., from 4 to 2 views).
4. How sensitive is HART to small pose misalignments or temporal inconsistencies (e.g., slight motion between views)? Could it handle unsynchronized or monocular sequences?
5. Despite being a feed-forward system, the Gaussian splatting stage still involves iterative optimization. Could a fully feed-forward rendering head be learned to replace this step for true real-time NVS?

**Questions:**

1. How does the model perform when trained or fine-tuned on real-world multi-view captures? Are there domain adaptation or photometric normalization strategies considered?

2. Besides the existing modules, what new representational capability does HART itself introduce? Could the same gains be achieved by tuning VGGT + DPSR with human-specific priors?

3. Could a fully feed-forward rendering head be learned to replace this step for true real-time NVS?

---

> ### Author Response · Authors · 2025-11-26
> **W1: Real-world training**
>
> We thank the reviewer for their thoughtful assessment and for highlighting the strengths of our work, in particular the integration of VGGT-based multi-view fusion with per-pixel SMPL-X attribute prediction as a human-specific adaptation of general 3D transformers, and the clear, reproducible presentation of our paper. Below, we address the main concerns raised: (W1) how HART behaves when trained or fine-tuned on real-world multi-view data, (W2) what new representational capability HART introduces beyond simply combining VGGT and DPSR, (W3) sensitivity to loss weights and robustness when reducing the number of input views, (W4) robustness to pose misalignment and temporal inconsistencies between views, and (W5) whether our rendering part could be feed-forward.
>
> **W1: Real-world training**
>
> We thank the reviewer for raising this important question. To the best of our knowledge, 4D-Dress [1] is the only real-world multi-view human dataset that provides both accurate clothed meshes and SMPL(-X) annotations, making it suitable for both training and quantitative evaluation. However, it contains far fewer distinct subjects than synthetic datasets (32 subjects, each with a tight and a loose outfit).
>
> To evaluate how HART behaves when trained with real-world data, we conduct an additional experiment where we mix synthetic THuman 2.1 and real 4D-Dress during training. We hold out 2 subjects (4 outfits) from 4D-Dress as a real-world test set, and use the remaining 30 subjects / 60 outfits for training. Following VGGT, we mix synthetic and real data with a 10:1 ratio to avoid overfitting to the limited number of real subjects, and re-train our model for 20 epochs (11k steps per epoch). At test time, we (i) randomly sample 100 frames from the held-out 4D-Dress subjects (stride ≥ 5) and (ii) evaluate cross-domain generalization on the 2K2K synthetic test set (which is never used for training).
>
> The quantitative results are as follows:
>
> | Training Dataset        |        4D-Dress        |        |        |        |        |        2K2K         |        |        |        |        |
> |------------------------|----------------------:|-------:|-------:|-------:|-------:|-------------------:|-------:|-------:|-------:|-------:|
> |                        | Acc. ↓                | Comp. ↓| CD ↓   | F-Score ↑ | NC ↑ | Acc. ↓             | Comp. ↓| CD ↓   | F-Score ↑ | NC ↑ |
> | THuman 2.1             | 0.0208                | 0.0274 | 0.0482 | 0.7504  | 0.8822 | 0.0077             | 0.0093 | 0.0170 | 0.9301  | 0.9479 |
> | **THuman 2.1 + 4D-Dress** | **0.0085**            | **0.0125** | **0.0210** | **0.9029** | **0.9186** | **0.0072** | **0.0089** | **0.0161** | **0.9348** | **0.9498** |
>
> | Training Set         |     4D-Dress       |         |        |        2K2K        |        |
> |---------------------|------------------:|--------:|-------:|------------------:|-------:|
> |                     | PA-V2V ↓          | PA-MPJPE ↓ |      | PA-V2V ↓          | PA-MPJPE ↓ |
> | THuman 2.1          | 21.60             | 22.93   |       | 22.86             | 24.49   |
> | **THuman 2.1 + 4D-Dress** | **19.09**         | **18.07** |   | **22.20**         | **23.10** |
>
> We find that incorporating 4D-Dress into the training set significantly improves both clothed-mesh and SMPL-X reconstruction metrics on the 4D-Dress test set. *This gain mainly comes from better handling of real-world effects such as strong boundary lighting*, which are not well represented in purely synthetic THuman 2.1 renders. In addition, adding real-world data slightly improves cross-domain performance on 2K2K (which is not used for training), suggesting that exposure to loose, diverse garments and complex clothing in the 4D-Dress training set helps our model generalize better to the challenging outfits in 2K2K. We also emphasize that even without any training on 4D-Dress, our original model already demonstrated strong generalization to loose clothing and human–object interactions on challenging datasets such as 2K2K and DNA-Rendering (as shown in Fig. 3 and Fig. 6). **For qualitative results on 4D-Dress, please refer to Fig. 12 in the updated paper (appendix)**.
>
> Regarding domain adaptation and photometric normalization, we do not employ any specialized adaptation techniques; our data preprocessing pipeline is essentially the same for both THuman 2.1 and 4D-Dress.
>
> **In addition, we also show an in-the-wild generalization by capturing and reconstructing a static person with a hand-held phone in Fig. 14 of the updated appendix.**

---

> > ### Author Response · Authors · 2025-11-26
> > **W4: Robustness to small pose misalignments or temporal inconsistencies**
> >
> > **W4: Robustness to small pose misalignments or temporal inconsistencies**
> >
> > We thank the reviewer for raising this point. We explicitly tested HART under such temporal inconsistencies on the two held-out test subjects in the 4D-Dress dataset. Specifically, we construct inputs where each of the 4 views comes from a different but nearby video frame in time, so that every view sees a slightly different pose. Despite these inter-view pose jitters, HART still reconstructs coherent, watertight clothed meshes without significant performance degradation, indicating that our method is robust to modest pose misalignments and small temporal motion between views. **Please refer to Fig. 13 in the updated appendix**.
> >
> > At the same time, our current formulation assumes that all images depict approximately the same pose. Large motions or long unsynchronized time gaps would eventually violate this assumption and are outside our current scope. Similarly, fully monocular dynamic sequences (where both pose and topology change over time) are not directly handled in this work; extending HART with temporal modeling to support true monocular video is an exciting direction for future research.

---

> ### Author Response · Authors · 2025-11-26
> **W2: Concern about novelty and differences between our method and VGGT+DPSR**
>
> **W2: Concern about novelty and differences between our method and VGGT+DPSR**
>
> HART introduces a human-specific, occlusion-aware volumetric representation on top of VGGT pointmaps, rather than just reusing DPSR as-is. While Shape As Points [4] proposes the differentiable Poisson solver and the idea of indicator–grid-level supervision, their setting focuses on point-cloud–to–mesh: they densify point clouds with MLPs and predict normals for generic scenes. In contrast, HART tackles a multi-view–to–human-geometry problem and defines a 3D U-Net that operates directly on partial indicator grids predicted from sparse views, explicitly designed to complete self-occluded human regions which are not well-observed in the input images (e.g., occluded limbs as shown in Fig. 3 of the main paper). This occlusion-aware refinement of the indicator field is new and is precisely what allows HART to “fill in” geometry where VGGT-style per-view pointmaps alone remain incomplete.
>
> Regarding the question of whether simply fine-tuning VGGT + DPSR would suffice, we explicitly test this in our ablations. We add a variant “Ours w/o Sapiens, w/o indicator grid refinement”, which is essentially VGGT + DPSR trained on our human data (no Sapiens-based normal refinement, no indicator grid refinement). Following reviewer PdSb’s suggestion, we report mean ± standard error of the mean (SEM) over 100 THuman 2.1 test subjects (4 random views each) to quantify uncertainty of the estimated mean:
>
> | Method                                           | Acc. ↓            | Comp. ↓           | CD ↓             | F-Score ↑        | NC ↑            |
> |--------------------------------------------------|------------------:|------------------:|-----------------:|-----------------:|----------------:|
> | Ours w/o Sapiens, w/o indicator grid refinement  | 0.0091±0.0005     | 0.0144±0.0010     | 0.0235±0.0015    | 0.9083±0.0035    | 0.8970±0.0025   |
> | Ours w/o Sapiens                                 | 0.0073±0.0003     | 0.0114±0.0008     | 0.0187±0.0010    | 0.9272±0.0032    | 0.9071±0.0023   |
> | Ours w/o indicator grid refinement               | 0.0087±0.0006     | 0.0138±0.0010     | 0.0224±0.0014     | 0.9156±0.0035    | 0.9022±0.0024   |
> | **Ours (full)**                                  | **0.0067±0.0003** | **0.0104±0.0007** | **0.0171±0.0009**| **0.9360±0.0032**| **0.9125±0.0022**|
>
> We observe that:
> - The VGGT + DPSR–like variant (w/o Sapiens, w/o indicator grid) is consistently the worst across all metrics.
> - **Adding either Sapiens-based residual normals or our indicator grid refinement yields monotonic improvements over this baseline**.
> - **The full HART model, combining both human-specific residual normal learning and occlusion-aware indicator grid refinement, achieves the best performance on all metrics**, with mean ± SEM intervals clearly separated from the ablated variants for most metrics.
>
> This ablation directly answers the reviewer’s concern: **simply tuning VGGT + DPSR with human data is not enough**. The occlusion-aware indicator grid 3D U-Net and a human-specific residual normal head trained end-to-end with DPSR are crucial to obtaining the reported gains in accuracy and completeness, and cannot be reproduced by a straightforward VGGT + DPSR fine-tuning.

---

> ### Author Response · Authors · 2025-11-26
> **W3: Lack of sensitivity/robustness analysis on loss weights and view number reduction**
>
> **W3: Lack of sensitivity/robustness analysis on loss weights and view number reduction**
>
> For loss weights, we largely inherit the design from VGGT and ETCH and deliberately avoid extensive tuning, as fully sweeping all combinations at this model scale would be prohibitively expensive. Concretely, for the four SMPL-related loss terms, we follow ETCH and set all weights to 1.0. For the four main objectives (pointmap, normal, DPSR, SMPL), we also set all weights to 1.0, following the simple, balanced weighting strategy used in VGGT.
>
> Nevertheless, we conducted a small sensitivity analysis. We trained two additional variants for 8 epochs (≈40% of the full schedule): one with DPSR loss weight = 2.0, and one with SMPL loss weight = 2.0, and compared test losses at epoch 8:
>
> | Setting            | DPSR loss ↓ | SMPL loss ↓ |
> |------------------|------------:|------------:|
> | Ours (default)    | 0.0009      | 0.6688      |
> | 2.0× DPSR weight  | 0.0010      | 0.6761      |
> | 2.0× SMPL weight  | 0.0009      | 0.6638      |
>
> We find that upweighting DPSR does not yield better refined indicator grids, and upweighting SMPL does not lead to significantly smaller SMPL losses on the test set either. This suggests that *our simple, balanced weighting is already in a reasonable regime and that aggressively increasing individual loss weights does not bring clear benefits*.
>
> Regarding robustness to view reduction, in our original appendix (Tab. 5), we already analyzed performances with 3, 4, 6, and 8 views. Following the reviewer’s request, we additionally report 2-view results and, following reviewer D7TP’s suggestion, we compare to the SOTA single-view method PSHuman [3] to explicitly test whether more views lead to better geometry:
>
> | Method            | Acc. ↓   | Comp. ↓  | CD ↓     | F-Score ↑ | NC ↑     |
> |------------------|---------:|---------:|---------:|----------:|---------:|
> | PSHuman (1 view) | 0.1862   | 0.4103   | 0.5965   | 0.3649    | 0.7407   |
> | Ours (2 views)   | 0.0420   | 0.0630   | 0.1050   | 0.6420    | 0.8609   |
> | Ours (3 views)   | 0.0088   | 0.0130   | 0.0218   | 0.9041    | 0.9057   |
> | Ours (4 views) | 0.0067 | 0.0105 | 0.0172 | 0.9354 | 0.9125 |
>
> Although our model is never trained on 2-view settings, we find that it’s still able to produce reasonable meshes. As expected, reducing the number of views from 4→3→2 causes a degradation in all geometry metrics, but the model does not catastrophically fail. Importantly, even with 2 views, our method still substantially outperforms the single-view PSHuman baseline across all metrics. **Please refer to Fig. 11 in the revised paper (appendix) for qualitative comparisons**.

---

> ### Author Response · Authors · 2025-11-26
> **W5: Rendering is not feed-forward**
>
> **W5: Rendering is not feed-forward**
>
> We agree that a fully feed-forward rendering head is an exciting direction, but it is orthogonal to the main focus of HART, which is unified feed-forward geometry (clothed mesh + SMPL-X) from sparse uncalibrated views. In this work, we therefore adopt a Gaussian splatting optimization stage initialized from our predicted clothed mesh, which already yields strong NVS performance with moderate training cost.
>
> Very recent concurrent works, such as Forge4D [7], explore exactly this direction by leveraging the VGGT backbone and learning feed-forward deformable 3D Gaussian splats that enable fast dynamic human novel-view synthesis from sparse-view videos. We view such approaches as highly complementary to HART: our unified human reconstruction module could serve as a strong geometric prior or initialization for future fully feed-forward NVS pipelines (e.g., in the spirit of Forge4D), and combining these two lines of work is an exciting avenue for future research.

---

### Official Review · Reviewer_veT3 · 2025-10-31

**Soundness:** 2
**Presentation:** 3
**Contribution:** 2
**Rating:** 4
**Confidence:** 4

**Summary:**

The paper introduces HART, a transformer-based framework for sparse-view human reconstruction (clothed mesh, SMPL-X estimation, novel-view synthesis) with occlusion-aware DPSR and residual normal learning.

**Strengths:**

1. One strength of HART is its unified transformer-based framework, which jointly outputs watertight clothed meshes, aligned SMPL-X body meshes, and Gaussian-splat representations for novel-view rendering from sparse uncalibrated RGB images—avoiding fragmented workflows of prior methods that handle these tasks separately. It achieves SOTA results (e.g., 18–23% CD improvement in clothed mesh reconstruction) across datasets like THuman 2.1 and DNA-Rendering, even with training on only 2.3K synthetic scans .

2. Another strength lies in its practical optimizations for human-specific challenges: it uses occlusion-aware DPSR with a 3D U-Net to recover complete geometry in self-occluded regions (a limitation of general 3D reconstruction backbones like VGGT) and leverages residual normal learning (with Sapiens priors) to enhance surface detail. These tweaks, though built on existing modules, better adapt feed-forward transformers to sparse-view human reconstruction scenarios .

**Weaknesses:**

1. HART’s novelty is not in new conceptual ideas, but in systematically integrating and adapting existing modules for sparse-view human reconstruction: it combines transformer architecture, occlusion-aware DPSR, and residual normal learning (no new network/reconstruction paradigm); applies Gaussian splatting (mainstream rendering) with human mesh constraints (no new rendering method); tailors pre-trained VGGT/DINOv2 (for general tasks) to human-specific tasks (no new extractors/encoders), focusing on cohesive pipeline optimization.

2. The method heavily relies on external components: removing the Sapiens model (for base normal prediction) results in blurrier surfaces, and disabling the indicator grid refinement leads to incomplete geometry in self-occluded regions, showing its weak independence from these specific auxiliary modules

3. HART is trained solely on 2,345 synthetic scans from the THuman 2.1 dataset, which lacks the diversity of real-world human variations (e.g., plus-size bodies, sheer fabrics, heavy coats, or diverse ethnic features). This limits the validity of claims about "robust generalization to real-world settings," as the training data does not fully represent real-world complexity.

4. While HART uses RANSAC and PnP to estimate camera parameters (instead of assuming centered principal points), the paper does not quantify the accuracy of these estimated parameters (e.g., comparison with ground-truth camera poses on test datasets). This omission hides potential biases—if camera estimation is inaccurate under certain conditions (e.g., extreme poses), it could confound the evaluation of reconstruction/rendering performance.

**Questions:**

Please refer to the weakness part.

**Details Of Ethics Concerns:**

N/A.

---

> ### Author Response · Authors · 2025-11-26
> **W1: Concern about novelty**
>
> We thank the reviewer for their careful reading and for recognizing the value of a unified transformer framework tailored to sparse-view human reconstruction and our effective adaptation of feed-forward transformers on human-specific challenges. Below, we address the main concerns raised: (W1) the level and nature of HART’s novelty beyond integrating existing modules, (W2) our reliance on components such as Sapiens and DPSR and how they are tightly integrated into a human-centric, end-to-end design, (W3) the scope and evidence behind our real-world generalization claims and the additional real-world training experiments, and (W4) the accuracy of our PnP-based camera estimation.
>
> **W1: Concern about novelty**
> We appreciate the reviewer’s detailed analysis and agree that HART builds on several existing components (VGGT backbone, Sapiens normals, DPSR-style reconstruction, Gaussian splatting). Our novelty, however, is not a simple plug-and-play integration, but lies in (1) adapting a VGGT-style transformer to human-specific, self-occluded geometry, and (2) designing a unified feed-forward framework that tightly couples clothed mesh reconstruction and SMPL-X estimation, and drives 2D Gaussian surfels for novel-view synthesis.
>
> Concretely, our contributions beyond prior work are:
> - **Human-centric extension of VGGT that resolves self-occlusion**. Vanilla VGGT is designed for generic scenes and predicts partial pointmaps that often *miss occluded regions*. In HART, **we introduce an occlusion-aware DPSR module** (indicator grid + 3D U-Net refinement) tailored to articulated humans, which fills in large missing regions (e.g., occluded limbs such as subject 1 in Fig. 3) and produces **accurate watertight meshes** that VGGT alone cannot reconstruct. This ability to complete human geometry under strong self-occlusion is not present in the original backbone and is central to our method.
> - **Unified feed-forward framework for coupled human reconstruction tasks**. Unlike previous approaches that require separate frameworks for clothed mesh reconstruction, SMPL estimation, and novel view synthesis, HART uses a single transformer that, from the same sparse uncalibrated inputs, simultaneously predicts (i) detailed clothed geometry and (ii) aligned SMPL-X body meshes via tightness vectors and part labels. The predicted clothed mesh then serves as a strong geometric prior to initialize a 2DGS representation, which is subsequently optimized for novel-view rendering, thus tightly coupling reconstruction, body model registration, and rendering within one coherent pipeline.
>
> In summary, while we intentionally reuse proven building blocks (transformers, Sapiens normal predictor, DPSR, Gaussian splatting) for robustness, **HART introduces a human-centric, occlusion-aware formulation and a unified multi-output architecture that together enable capabilities that prior methods — including VGGT itself — do not provide: complete clothed meshes plus aligned SMPL-X from sparse uncalibrated views in a single forward pass**, along with high-quality novel view synthesis initialized from the predicted clothed geometry. We view our method as a concrete step toward **holistic and efficient human reconstruction systems that can scale to casual, uncalibrated, in-the-wild multi-view captures, where accurate geometry, pose, and rendering can be produced jointly** and efficiently rather than via separate pipelines.

---

> ### Author Response · Authors · 2025-11-26
> **W2: Concern about reliance on external components**
>
> **W2: Concern about reliance on external components**
>
> We agree that HART leverages strong existing components where appropriate, but we would like to clarify that these modules are not used as simple plug-ins; they are tightly integrated into our design. First, unlike generic dense-prediction work, **our residual normal learning is tightly coupled with DPSR in an end-to-end differentiable manner**. The Sapiens-based prior, residual normal head, and indicator grid refinement are co-designed and trained jointly so that improvements in predicted normals directly translate into better clothed human geometry, as confirmed by our ablations.
>
> Second, the indicator grid refinement is not a component borrowed wholesale from prior work; it is a key part of our contribution. We build on the DPSR formulation of Shape As Points [4], which introduces a differentiable Poisson solver and indicator-grid-level supervision, but their setting is fundamentally different: they densify input point clouds with MLPs and predict normals for generic point-cloud–to–mesh reconstruction. In contrast, HART addresses a multi-view–to–human-geometry problem, and we design a **human-specific 3D U-Net operating directly on partial indicator grids, explicitly tailored to fill in self-occluded regions**. This occlusion-aware refinement of the indicator field is new and is central to why we can complete self-occluded regions where VGGT-style per-view pointmaps fail.
>
> Thus, rather than indicating “weak independence”, our ablation results show that these components provide systematic, monotonic gains: the baseline VGGT+DPSR pipeline already works, but **adding Sapiens-based residual normals and our indicator grid refinement consistently improve all metrics of the reconstructed human geometry**.

---

> ### Author Response · Authors · 2025-11-26
> **W3: Concern on real-world generalization**
>
> **W3: Concern on real-world generalization**
>
> We agree that THuman 2.1 alone does not cover the full spectrum of real-world human variation (plus-size bodies, very heavy outerwear, all fabric types, etc.). Our claim is instead empirically grounded in the fact that, despite training only on 2.3K synthetic THuman 2.1 scans, **HART generalizes well to challenging out-of-domain human datasets such as 2K2K [5] and DNA-Rendering [6]**, which contain significantly more diverse identities, clothing, and human–object interactions than our training data.
>
> In particular, *DNA-Rendering is a real-world multi-view dataset explicitly designed to stress-test novel view synthesis with loose, diverse garments and rich human–object interactions*. Even though **our HART transformer is never trained or fine-tuned on DNA-Rendering**, it achieves state-of-the-art novel-view synthesis performance on this dataset (Tab. 3, Figs. 5 and 7) and produces qualitatively accurate meshes for these challenging subjects (Fig. 6).
>
> Following reviewer iAZE’s suggestion, we also conduct an additional real-world training experiment and show that HART can further benefit from extra real-world supervision. Specifically, we mix THuman 2.1 with the 4D-Dress [1] dataset (the only real-world multi-view dataset with both clothed meshes and SMPL(-X) annotations), using a 10:1 synthetic–real sampling ratio. **As reported in Tabs. 12–13 and Fig. 12 in the updated appendix, this mixed training significantly improves both clothed-mesh and SMPL-X metrics on the held-out 4D-Dress test set, and slightly improves cross-domain performance on 2K2K (which remains unseen during training)**. Together, these results support our generalization claim: **even when trained only on synthetic data, HART generalizes well to substantially different real-world benchmarks, and continues to improve as more real-world data becomes available**.
>
> **In addition, we also show an in-the-wild generalization by capturing and reconstructing a static person with a hand-held phone in Fig. 14 of the updated appendix**.

---

> ### Author Response · Authors · 2025-11-26
> **W4: Concern about camera estimation**
>
> **W4: Concern about camera estimation**
>
> We fully agree that the quality of the estimated cameras is critical, since these poses are used both in DPSR and in Gaussian splatting, and inaccurate cameras could in principle bias the reconstruction and rendering evaluation. We did not originally report camera metrics because camera estimation is not a core contribution of HART, and we simply adopt a standard pipeline introduced in Dust3R [2] and later used in many follow-up works: focal lengths are recovered with the Weiszfeld algorithm, and extrinsics are obtained via PnP-RANSAC on dense pointmaps.
>
> Although our backbone is VGGT, we deliberately do not use its feed-forward camera head, because it is pretrained under the assumption of centered principal points, which is reasonable for many scene-level datasets but not for our foreground human setting where the subject is often off-center. In our pipeline, we always recenter the human bounding box to maximize foreground coverage and improve training stability, which effectively changes the principal point and makes the original VGGT camera head mismatched to our training data.
>
> To quantify how accurate our PnP-based cameras are and whether they could confound results, we perform a controlled comparison against the VGGT camera head on THuman 2.1. We re-render THuman 2.1 such that both the principal point and the human bounding box are centered: we keep the principal point fixed at the image center and shift camera translations in 3D so that the person remains centered in 2D. We then retrain our method and finetune VGGT with its camera head on this dataset and evaluate on two test splits with the same 100 subjects:
> - Test set 1 (centered principal points): in-domain test set where both the human bounding boxes and the ground-truth principal point are at the image center.
> - Test set 2 (off-center principal points / centered human bounding boxes): our original THuman 2.1 test setting which is cross-domain for the new model, where we center the human bounding box at inference (more representative of our real usage), so that ground-truth intrinsics are off-centered.
>
> We report (i) R/T pose AUC@30 on Test set 1, following VGGT, and (ii) 2D reprojection AUC@10px, which measures how well the predicted pointmaps project onto 2D align with the GT foreground pixels and is most relevant for downstream novel view synthesis. We only report R/T AUC@30 on Test set 1, where ground-truth principal points are centered and the pose metric is well-defined; on Test set 2, ground-truth intrinsics are off-centered while our estimated intrinsics are constrained to have centered principal points, so pose error alone is not diagnostic of camera accuracy. The results are as follows:
>
> | Methods            | R/T AUC@30 ↑ (Centered PP) | 2D Reproj. AUC@10px ↑ (Centered PP) | 2D Reproj. AUC@10px ↑ (Centered Human BBox) |
> |-------------------|---------------------------:|------------------------------------:|--------------------------------------------:|
> | VGGT camera head  | **99.97**                  | 97.73                               | 12.13                                       |
> | Ours              | 96.29                      | **99.49**                           | **95.28**                                   |
>
> These experiments show that our PnP-based cameras are on par with VGGT in terms of pose accuracy under the ideal, in-distribution centered principal points setting (Test set 1), while providing much better 2D alignment when evaluated under the more realistic off-center regime (Test set 2). In contrast, VGGT camera head overfits to the image distribution that have centered principal points, and when tested on out-of-distribution off-center settings, produces significantly misaligned 2D projections. Since accurate 2D reprojection is what most directly impacts Gaussian splatting, our choice of the PnP-RANSAC camera estimation actually reduces error accumulation in the following rendering stage.

---

### Official Review · Reviewer_PdSb · 2025-10-31

**Soundness:** 3
**Presentation:** 3
**Contribution:** 3
**Rating:** 6
**Confidence:** 4

**Summary:**

The paper introduces a feed-forward transformer that takes a set of >=3 uncalibrated RGB images as input, and produces a watertight clothed mesh, aligned SMPL body mesh, and 3DGS representation for photorealistic novel-view rendering. The transformer design is initialized from VGGT, with DPT heads for residual normals (on top of Sapiens), pointmaps (which are combined with normal predictions for differentiable Poisson surface reconstruction and clothed mesh reconstruction), and SMPL tightness vectors (as in ETCH) as well as body part label maps (which are together decoded into SMPL-X meshes). The method is trained on 2.3k human meshes from THuman2.1 dataset. 2D Gaussian surfels can be optionally outputted for novel-view synthesis.

**Strengths:**

- Developing feed-forward VGGT-style architectures for clothed human reconstruction from multiple views is an important and interesting problem.
- The method is straightforward and well-motivated. The key introduced components (normal residual prediction and indicator grid refinement) help improve the quality of the geometry according to an ablation study
- The method produces better results than baselines such as VGGT, MAtCha, PuzzleAvatar, LaRa, etc.

**Weaknesses:**

- The reconstructed meshes lack very fine-grained details, such as detailed clothing textures, fingers, and hair. In the conclusion, the authors claim this is due to limited indicator grid resolution. Fundamentally, despite predicting residuals on top of a Sapiens normal prior, it seems the geometry quality is slightly better than VGGT but not substantially better.
- The impact of predicting residual normals on top of Sapiens and indicator grid refinement seems qualitatively very small, at least according to Figure 9 and Table 8. It is not clear to me whether the impact is significant.
- The method requires >=3 uncalibrated human images captured in the same body pose, which may be difficult to obtain in the wild outside of synchronized, sparse-view camera capture setups. Extending the method to dynamic monocular videos may be an interesting avenue for future research

**Questions:**

- Which dataset and sequences are the ablation studies in A.7 performed on?
- Regarding the statistical significance of the ablation studies in A.7, is it possible to show per-sequence ablation results or error bars with multiple randomly-selected view sets per scene, to confirm whether the proposed design choices yield consistent improvements on every sequence?

---

> ### Author Response · Authors · 2025-11-26
> **W1: Lack of fine-grained details**
>
> We thank the reviewer for their thoughtful comments and for highlighting the importance of developing feed-forward VGGT-style architectures for multi-view clothed human reconstruction and the role of our residual normal prediction and indicator-grid refinement in improving geometry quality. Below, we address the main points raised: (W1) the lack of extremely fine-grained details and how our core contribution lies in faithful, self-occlusion-aware geometry completion rather than hallucinated details, (W2) the quantitative and qualitative impact of Sapiens normals and indicator-grid refinement (including new normal-map evaluations), (W3) the practical use case of our ≥3-view setting and its relation to dynamic monocular video, and (Q1–Q2) clarification and strengthened statistical analysis of our ablations with mean ± SEM.
>
> **W1: Lack of fine-grained details**
>
> We agree that our current pipeline does not capture extremely fine, high-frequency details such as intricate clothing textures, individual hair strands, or fully articulated fingers, and we already acknowledge this as a limitation due to the memory–resolution trade-off in our indicator grid. However, both our quantitative results and visualizations show a consistent and meaningful improvement over VGGT: beyond slightly better local surface quality, our method fills in missing or self-occluded regions of geometry that VGGT simply fails to reconstruct (e.g., the occluded limbs of subject 1 in Fig. 3 of the main paper), producing a coherent, watertight clothed mesh rather than partial or hole-ridden surfaces. This ability to complete the geometry under strong self-occlusion is the core contribution of our method.
>
> We also emphasize that our goal is faithful reconstruction of the ground-truth geometry, not hallucination of extra fine-grained details. Single-view methods such as PSHuman [3] can synthesize sharper-looking details, but their mesh reconstructions are often not geometrically accurate (e.g., incorrect body poses and shapes). In contrast, our multi-view, feed-forward framework is designed to robustly recover accurate clothed human geometry and SMPL bodies from sparse, uncalibrated images, with the main gain over VGGT coming from complete, self-occlusion-aware geometry rather than purely from sharper textures. **Please refer to Fig. 11 in the updated appendix for qualitative comparisons between PSHuman and our method**.

---

> ### Author Response · Authors · 2025-11-26
> **W2: Unclear improvement from Sapiens normals and indicator grid refinement**
>
> **W2: Unclear improvement from Sapiens normals and indicator grid refinement**
>
> While the visual differences in Fig. 9 may appear subtle at a glance, both residual normal prediction and indicator grid refinement bring consistent quantitative improvements across all evaluated sequences and metrics. In particular, as shown in Tab. 8, removing either component leads to a consistent degradation in all geometric metrics, which indicates that their effect is systematic rather than incidental.
>
> Following reviewer D7TP’s request, we have also conducted an evaluation on the normal maps before and after refinements:
>
> | Methods                          | Cosine Similarity ↑ | L1 Loss ↓ |
> |----------------------------------|-------------------:|----------:|
> | Ours w/o Sapiens                 | 0.9548             | 0.1131    |
> | Sapiens Monocular Normals        | 0.9553             | 0.1148    |
> | **Ours**                         | **0.9671**         | **0.0951** |
>
> As these results show, the DPT-based normal refinement substantially improves both cosine similarity and L1 error over the ablated versions, indicating more accurate normals overall. In practice, we also observe that our refined normals provide sharper local details compared to ablated variants that predict normals from scratch, and better multi-view 3D consistency than the Sapiens monocular normals. **We have also included both before/after DPT normal visualizations (Fig. 10) in the revised paper (appendix)**.

---

> ### Author Response · Authors · 2025-11-26
> **W3: Use case of our method and extension to dynamic monocular videos**
>
> **W3: Use case of our method and extension to dynamic monocular videos**
>
> We thank the reviewer for this insightful suggestion. Extending our framework to dynamic monocular videos is indeed an exciting direction for future work.
>
> We would like to emphasize, however, that our current requirement of ≥3 uncalibrated images in a single pose is already more practical and scalable than most existing multi-view methods that rely on synchronized multi-camera rigs and pre-calibrated cameras in controlled lab environments. In our setting, the input can be acquired simply by walking around a static person with a handheld camera or phone; the cameras are not required to be calibrated beforehand, and our pipeline recovers camera parameters from the images themselves. **An in-the-wild example of this capture protocol is shown in Fig. 14 of the updated appendix**.
>
> Thanks to reviewer iAZE’s suggestion, we further observe that HART is robust to small pose and viewpoint jitter between the input frames: as long as the subject roughly maintains a pose, *small motion difference between views does not prevent accurate reconstruction*. **Please refer to Fig. 13 in the updated appendix for qualitative results**.
>
> However, our current formulation still assumes that all input images capture roughly the same static pose. If there are large motions or long time gaps between views, this assumption breaks and our method is not designed to handle those cases. Likewise, fully dynamic monocular videos, where both pose and even clothing topology evolve over time, are beyond the scope of this work; extending HART with explicit temporal modeling to tackle such truly monocular, time-varying sequences is an exciting direction for future research.

---

> ### Author Response · Authors · 2025-11-26
> **Q1: Which dataset and sequences are the ablation studies in A.7 performed on?**
>
> **Q1: Which dataset and sequences are the ablation studies in A.7 performed on?**
>
> The ablation studies in A.7 are performed on the THuman 2.1 test set with 100 unseen subjects, the same test set used for the THuman evaluations in the main paper. We have added a clarification in the revised version of the appendix.

---

> ### Author Response · Authors · 2025-11-26
> **Q2: Regarding the statistical significance of the ablation studies in A.7, is it possible to show per-sequence ablation results or error bars with multiple randomly-selected view sets per scene, to confirm whether the proposed design choices yield consistent improvements on every sequence?**
>
> **Q2: Regarding the statistical significance of the ablation studies in A.7, is it possible to show per-sequence ablation results or error bars with multiple randomly-selected view sets per scene, to confirm whether the proposed design choices yield consistent improvements on every sequence?**
>
> We thank the reviewer for the suggestion. We have recomputed the ablation studies on the 100 THuman 2.1 test subjects, each evaluated with 4 randomly selected test views. Following reviewer iAZE’s suggestion, we also include an additional ablation “Ours w/o sapiens, w/o indicator grid refinement”, which is effectively just VGGT + DPSR with human priors. For every entry, we now report the mean ± standard error of the mean (SEM)：
>
> | Method                                           | Acc. ↓            | Comp. ↓           | CD ↓             | F-Score ↑        | NC ↑            |
> |--------------------------------------------------|------------------:|------------------:|-----------------:|-----------------:|----------------:|
> | Ours w/o Sapiens, w/o indicator grid refinement  | 0.0091±0.0005     | 0.0144±0.0010     | 0.0235±0.0015    | 0.9083±0.0035    | 0.8970±0.0025   |
> | Ours w/o Sapiens                                 | 0.0073±0.0003     | 0.0114±0.0008     | 0.0187±0.0010    | 0.9272±0.0032    | 0.9071±0.0023   |
> | Ours w/o indicator grid refinement               | 0.0087±0.0006     | 0.0138±0.0010     | 0.0224±0.0014     | 0.9156±0.0035    | 0.9022±0.0024   |
> | **Ours (full)**                                  | **0.0067±0.0003** | **0.0104±0.0007** | **0.0171±0.0009**| **0.9360±0.0032**| **0.9125±0.0022**|
>
> As shown in the table, **our full model consistently outperforms all ablated variants across all metrics**. For ↑ metrics (F-Score, NC), the lower bound of our full model (mean − SEM) is already above the upper bound of each ablated variant (mean + SEM), while for ↓ metrics (Acc., Comp., CD), the upper bound of our model (mean + SEM) is mostly below the lower bound of the ablations (mean − SEM). This separation of mean ± SEM intervals indicates that the improvements from the Sapiens prior and indicator grid refinement are consistent and unlikely to be due to sampling noise.

---

### Official Review · Reviewer_D7TP · 2025-11-02

**Soundness:** 2
**Presentation:** 2
**Contribution:** 2
**Rating:** 4
**Confidence:** 4

**Summary:**

This paper proposes HART, a unified feed-forward transformer framework for sparse-view human reconstruction, which  simultaneously predicts clothed human geometry, body pose, and GS. The work builds upon  VGGT and extends it to human-centric reconstruction by introducing multiple prediction heads and an occlusion-aware differentiable Poisson reconstruction (DPSR) module. The experiment shown improvements against baselines in geometric, rendering and pose estimation.

**Strengths:**

- The transformer based feed-forward pipeline should be much higher efficient compared to optimization- or diffusion-based methods；
- It is reasonable to achieve various human-centric tasks, like pose, pointmap or geometry and potential tracking, in a unified model.

**Weaknesses:**

1. I think the claim of  “scalable model” in abstract is not proper, because hart is trained with 3d scans dataset.
2. The performance is not impressive for me, especially the geometry and smplx results. The human mesh in fIg.3 and video demos, are over-smoothed and on the same par with Vggt predictions. The challenges of  pose estimation tasks mainly lies in hand region and in-the-wild challenging cases. However, this work dose not preform very well.
3. I have great concerns about the normal predictions of DPT heads, even if it is designed for dense prediction tasks. It will be better to include some normal outputs   before/after DPT refinement.
4. Another concern is the accuracy of the PnP-predicted camera poses. It is critical, as the estimated poses are used in both DPSR  and GS fitting, where accumulated errors could degrade overall performance.
5. The comparisons are not sufficient. although this work focus on mv setting, the comparisons should include single view based methods, like SiTH and PSHuman. With more input views, this work should perform better than them.

**Questions:**

as listed in weakness.

---

> ### Author Response · Authors · 2025-11-26
> **W1: “Scalable model” claim**
>
> We thank the reviewer for their thoughtful feedback and for highlighting the value of a unified, feed-forward transformer pipeline for efficient human reconstruction from sparse views. Below, we address the main concerns raised in the review: (W1) what we mean by “scalable” and how HART benefits from additional real-world training data, (W2) the perceived lack of fine details and our true improvements over existing baselines in both geometry and SMPL-X, (W3) the reliability of our DPT-based normal refinements, (W4) the accuracy of our PnP-based camera estimation, and (W5) new experiments against the single-view PSHuman baseline.
>
> **W1: “Scalable model” claim**
>
> We thank the reviewer for pointing this out. Our intention with the term “scalable model” is to emphasize that HART is **a feed-forward transformer** that (1) performs feed-forward inference on a “test subject” instead of requiring per-subject optimization, and (2) can directly benefit from additional data as suitable datasets become available. Unlike optimization- or SDS-based pipelines that require minutes to hours of per-subject fitting, our model performs geometry reconstruction in a single forward pass, which makes it inherently more scalable in terms of computational cost and deployment (e.g., processing many subjects or many frames).
>
> The main limitation lies in the **data**, not in the model design. High-quality multi-view human datasets that provide both clothed meshes and accurate SMPL-X annotations are extremely scarce; to the best of our knowledge, THuman 2.1 is currently the only large-scale public dataset with this combination, which restricts our training set to 2.3K scans. Nevertheless, we already demonstrate strong cross-dataset and real-world generalization without any additional fine-tuning: as shown in Tab. 1–2 and Fig. 3–4, HART trained only on THuman 2.1 achieves state-of-the-art performance on 2K2K (with different subjects, clothing, and interactions). Furthermore, as shown in Tab. 3 and Fig. 5 of the main paper and Tab. 6 and Fig. 7 in the appendix, the same model attains state-of-the-art novel-view synthesis results on the real-world DNA-Rendering dataset, and produces accurate clothed mesh reconstructions there as illustrated in **Fig. 6 of the appendix**.
>
> To further validate the “scalable with more data” aspect, and according to the suggestion of reviewer iAZE, we additionally train HART on a mixture of THuman 2.1 + 4D-Dress [1], the only real-world multi-view dataset we are aware of that provides both clothed meshes and SMPL(-X) annotations. We find that incorporating 4D-Dress significantly improves both clothed-mesh and SMPL-X metrics on the 4D-Dress test set, and also slightly improves cross-domain performance on 2K2K (which remains unseen during training). This confirms that, as more real-world multi-view data becomes available, HART can directly absorb it and improve, consistent with our intended notion of scalability. **Please refer to Tab. 12-13 and Fig. 12 in the updated paper (appendix) for quantitative and qualitative evaluations between our models trained with or without real-world data**.
>
> We have clarified in the abstract accordingly: we explicitly state that “scalable” refers to the feed-forward nature and data-efficiency of the framework, i.e., its ability to perform per-subject human geometry reconstruction without optimization and to improve as more training data is added, rather than to the current size or diversity of existing human datasets, which is a limitation of available benchmarks rather than of our model.

---

> > ### Author Response · Authors · 2025-11-26
> > **W2: Lack of fine-grained details and concern on real-world generalization**
> >
> > **W2: Lack of fine-grained details and concern on real-world generalization**
> >
> > We appreciate the reviewer’s observation about the lack of very fine-scale details (e.g., fingers, hair) and agree that this is a limitation of our current pipeline. This is primarily due to the limited resolution of the indicator grid used in our DPSR module (512x512x512): increasing the grid resolution further quickly leads to prohibitive GPU memory usage and out-of-memory failures during training, so we necessarily trade some fine details for a feasible memory footprint. In future work, we plan to explore incorporating multi-scale representations to better capture such high-frequency geometry.
> >
> > That said, we would like to emphasize that, even with this limitation, HART already outperforms existing methods on clothed mesh reconstruction both qualitatively and quantitatively (Fig. 3 and Tab. 1). Among the baselines,
> > - **VGGT** struggles to recover self-occluded regions;
> > - **PuzzleAvatar** benefits from SMPL initialization for facial details but fails on loose garments and human–object interactions;
> > - **MAtCha** yields overly noisy surfaces with holes; and
> > - **LaRa** produces overly smoothed geometry.
> >
> > As recommended by the reviewer, we additionally conduct qualitative and quantitative comparisons against PSHuman (see W5). While PSHuman can recover fine details, such as fingers using a high-resolution multi-view diffusion model, these details are often hallucinated and not faithful to the ground-truth geometry, which is reflected in significantly worse reconstruction metrics. Designing a method that simultaneously preserves such fine-grained details and faithfully recovers the true clothed geometry remains an open challenge for future work.
> >
> > Similarly, we do not specifically target face/hand reconstruction accuracy for SMPL-X reconstruction: following ETCH, our body marker set does not include facial or finger keypoints, and our focus is on improving full-body SMPL-X pose and shape from multi-view images rather than detailed hand poses. Even so, our method still substantially outperforms all baselines on both in-domain and cross-dataset evaluations for SMPL-X estimation (Fig. 4 and Tab. 2), despite not modeling the finest details.
> >
> > Regarding challenging poses and in-the-wild generalization, we show some qualitative results for clothed mesh reconstruction and SMPL-X estimation on the 4D-Dress test set including some challenging poses. Our method is capable of reconstructing those poses accurately. **Please refer to the new qualitative comparisons in Fig. 12 of the revised paper (appendix). In addition, we also show an in-the-wild generalization by capturing and reconstructing a static person with a hand-held phone in Fig. 14 of the updated appendix**.
> >
> > Finally, we would like to emphasize that the main advantage of our method over VGGT does not lie in significantly better details, but in handling self-occlusions that are not well-observed in any of the input views: by explicitly modeling self-occluded regions with the indicator grid refinement, HART reconstructs self-occluded regions that VGGT cannot inherently model **(e.g., the occluded limbs of subject 1 in Fig. 3)**.

---

> ### Author Response · Authors · 2025-11-26
> **W3: Concern about normal DPT**
>
> **W3: Concern about normal DPT**
>
> In the original appendix, we included an ablation study (Tab. 8) to show the impact of leveraging Sapiens normals on the final clothed mesh reconstructions. To address the concern of the direct impact of DPT-based normal refinement, we have conducted additional quantitative and qualitative evaluations of our normal map predictions. We compare three variants against ground-truth normals using per-pixel cosine similarity and L1 error:
> | Methods                          | Cosine Similarity ↑ | L1 Loss ↓ |
> |----------------------------------|-------------------:|----------:|
> | Ours w/o Sapiens                 | 0.9548             | 0.1131    |
> | Sapiens Monocular Normals        | 0.9553             | 0.1148    |
> | **Ours**                         | **0.9671**         | **0.0951** |
>
> As these results show, the DPT-based normal refinement substantially improves both cosine similarity and L1 error over the ablated versions (predicting normals from scratch or just using Sapiens normals without refinement). In practice, we also observe that our refined normals provide sharper local details compared to ablated variants that predict normals from scratch, and better multi-view 3D consistency than the Sapiens monocular normals. We have also included both before/after DPT normal visualizations and the above quantitative table in the revised paper to make this clearer. **Please refer to qualitative comparisons in Fig. 10 of the revised paper (appendix)**.

---

> > ### Author Response · Authors · 2025-11-26
> > **W4: Concern about camera estimation**
> >
> > **W4: Concern about camera estimation**
> >
> > We agree that camera accuracy is important, since the estimated poses are used in both DPSR and GS fitting. We did not originally report camera metrics because camera estimation is not a core contribution of HART, and we follow a standard pipeline introduced in Dust3R [2] and adopted by many of its follow-up works: focal lengths are recovered with the Weiszfeld algorithm, and extrinsics are recovered by PnP-RANSAC applied to dense pointmaps.
> >
> > Although our backbone is VGGT, we intentionally avoid using its feed-forward camera head because it is pretrained assuming centered principal points, which holds for many scene-level datasets but not for human datasets where the foreground subject is often off-center. In our setting, we need to center the human bounding box to maximize foreground coverage and improve training stability; this changes the effective principal point, making the original VGGT camera head not suitable for our data.
> >
> > To directly compare our PnP-based cameras with VGGT’s camera head, we re-render THuman 2.1 so that both the principal point and the human bounding box are centered: we keep the principal point fixed at the image center, and shift the camera translations in 3D to compensate for centering the human bounding box in 2D. We retrain both methods on this controlled dataset. We then evaluate on two test splits with the same 100 subjects:
> > - Test set 1 (centered principal points): in-domain test set where both the human bounding boxes and the GT principal points are centered.
> > - Test set 2 (off-center principal points / centered human bounding boxes): our original THuman 2.1 test set which is cross-domain for the new model, where we re-center the human bounding box at inference (more aligned with real-world inputs), so that the ground-truth principal points are off-centered.
> >
> > We report (i) R/T pose AUC@30 on test set 1, following VGGT, to measure the extrinsics accuracy, and (ii) 2D reprojection AUC@10px, which measures how well the 2D projections of predicted pointmaps align with the GT foreground pixels, which is critical for the subsequent novel view synthesis task. We only report R/T AUC@30 on test set 1, where the ground-truth principal points are centered and this metric is well-defined; on test set 2, the ground-truth intrinsics are off-centered while our estimated intrinsics are constrained to have centered principal points, so pose error alone is not diagnostic of camera accuracy.
> >
> > | Methods            | R/T AUC@30 ↑ (Centered PP) | 2D Reproj. AUC@10px ↑ (Centered PP) | 2D Reproj. AUC@10px ↑ (Centered Human BBox) |
> > |-------------------|---------------------------:|------------------------------------:|--------------------------------------------:|
> > | VGGT camera head  | **99.97**                  | 97.73                               | 12.13                                       |
> > | Ours              | 96.29                      | **99.49**                           | **95.28**                                   |
> >
> >
> >
> >
> >
> >
> > These results show that our PnP-based cameras are on par with VGGT on the ideal and in-distribution centered setting, while providing dramatically better 2D alignment when evaluated under out-of-distribution and off-center settings that are closer to real-world settings. VGGT camera head, in contrast, is overfitted to centered-principal-points image distributions and produces significantly misaligned 2D projections when tested on off-center settings. Since accurate 2D reprojection is what most directly affects novel view synthesis, our choice of the PnP-RANSAC camera estimation actually reduces error accumulation in the downstream tasks.

---

> > > ### Author Response · Authors · 2025-11-26
> > > **W5: Lack of single-view baselines**
> > >
> > > **W5: Lack of single-view baselines**
> > >
> > > We thank the reviewer for this suggestion and have added a comparison to the state-of-the-art single-view clothed human reconstruction method PSHuman [3]. We directly evaluate PSHuman using its released checkpoint, as it has already been trained on the same THuman 2.1 dataset. For each test subject, we use a single front view as input to PSHuman and align its prediction to the ground-truth mesh using ICP before evaluation. Since our method is not designed for single-view hallucination (we do not adopt a diffusion prior to infer unseen geometry), we report results with 2, 3, and 4 views, always including the front view. Although our model is only trained with 3-8 views inputs, it still generalizes to 2-view settings. The results are as follows:
> > >
> > > | Method            | Acc. ↓   | Comp. ↓  | CD ↓     | F-Score ↑ | NC ↑     |
> > > |------------------|---------:|---------:|---------:|----------:|---------:|
> > > | PSHuman (1 view) | 0.1862   | 0.4103   | 0.5965   | 0.3649    | 0.7407   |
> > > | Ours (2 views)   | 0.0420   | 0.0630   | 0.1050   | 0.6420    | 0.8609   |
> > > | Ours (3 views)   | 0.0088   | 0.0130   | 0.0218   | 0.9041    | 0.9057   |
> > > | Ours (4 views) | 0.0067 | 0.0105 | 0.0172 | 0.9354 | 0.9125 |
> > >
> > > In practice, we observe that although PSHuman can hallucinate fine details with a high-resolution multi-view diffusion model, it often fails to faithfully reconstruct the true body pose and shape from a single image, leading to noticeable misalignment with the ground-truth mesh even after ICP alignment. In contrast, our method leverages multiple views to faithfully recover the actual geometry rather than hallucinating unseen regions. As expected, our performance improves as more views are provided: 2-view reconstructions (front + back) are already substantially better than PSHuman, and 3/4-view reconstructions further reduce Chamfer distances and increase F-Score and normal consistency. This directly confirms the reviewer’s hypothesis that, with more input views, our multi-view feed-forward approach does indeed outperform single-view methods such as PSHuman. **Please refer to Fig. 11 of the revised paper for qualitative results**.

---

### Author Response · Authors · 2025-11-26
**Summary of the author responses**

We sincerely thank all four reviewers for their careful evaluation, constructive suggestions, and for highlighting both the strengths and limitations of our work. Across the reviews, there is broad agreement that HART (i) provides an efficient and unified transformer framework that jointly predicts clothed meshes and underlying SMPL-X bodies (ii) extends VGGT with human-specific, occlusion-aware geometry completion, (iii) achieves state-of-the-art performance on multiple synthetic and real-world benchmarks (THuman 2.1, 2K2K, DNA-Rendering).

We would like to emphasize that HART’s conceptual novelty lies in

1. making VGGT **human-centric** and explicitly capable of handling self-occluded regions that are not visible in the input images, and in coupling this with a unified feed-forward architecture. Vanilla VGGT is designed for generic scenes and produces partial pointmaps that often miss heavily occluded regions. In HART, we introduce an **occlusion-aware DPSR module with a human-specific 3D U-Net for indicator grid refinement**, that operates directly on the partial predictions of VGGT backbone and is explicitly tailored to articulated humans. This allows us to complete large missing regions such as occluded limbs **(e.g., subject 1 in Fig. 3)** and to recover accurate, watertight clothed meshes that VGGT alone cannot reconstruct.

2. Additionally, our unified framework jointly predicts clothed geometry and aligned SMPL-X body attributes from the same sparse, uncalibrated inputs, and initializes a 2DGS representation for novel-view rendering.
We view HART as a concrete step toward holistic and efficient human reconstruction systems that can scale to casual, uncalibrated, in-the-wild multi-view captures, where accurate geometry, pose, and rendering are produced jointly and coherently rather than via separate, loosely coupled pipelines.

To respond to the reviewers’ concerns, we have added the following **7 new experiments**:
1. **Evaluation on normal maps** (reviewer D7TP): quantitative/qualitative comparisons on the normal map predictions with/without DPT refinement **(Fig. 10, Tab. 9)**.
2. **Evaluation on camera accuracy** (reviewer D7TP, veT3): conduct a controlled study comparing our PnP-RANSAC cameras with the VGGT camera head under centered and off-center principal-point settings **(Tab. 11)**.
3. **Comparison with single-view baseline for clothed mesh reconstruction** (reviewer D7TP): quantitative and qualitative comparisons to PSHuman, showing that our multi-view reconstructions are significantly more faithful to the ground-truth geometry than the single-view methods **(Fig. 11, Tab. 10)**.
4. **Ablation studies with uncertainties and an additional variant of VGGT+DPSR** (reviewer PdSb, iAZE): extend our ablations to include an “Ours w/o Sapiens, w/o indicator grid refinement’’ variant (effectively VGGT + DPSR), and report mean ± SEM showing that both of our proposed components lead to statistically meaningful gains **(Tab. 8)**.
5. **Training and evaluating on real-world datasets** (reviewer iAZE): while our original THuman-only model already generalizes well to DNA-Rendering without any real-world fine-tuning **(Fig. 6)**, we further demonstrate that mixing THuman 2.1 with the real-world 4D-Dress dataset for training significantly boosts clothed-mesh and SMPL-X accuracy on the 4D-Dress test set and slightly improves cross-domain performance on the 2K2K test set **(Tab. 12-13, Fig. 12)**.
6. **Evaluation on robustness to pose misalignments/temporal inconsistencies** (reviewer iAZE): conduct experiments where each view comes from a different but nearby video frame in time, and show that HART still produces coherent, watertight reconstructions, indicating robustness to modest pose misalignments **(Fig. 13)**.
7. **In-the-wild generalization** (reviewer D7TP, PdSb, veT3, iAZE): We evaluate our method on a casual in-the-wild capture, where a static subject is recorded using a handheld phone while the user walks around them. Our model robustly reconstructs a high-fidelity clothed mesh along with the aligned SMPL-X body mesh in a single feed-forward pass **(Fig. 14)**.

In addition, we have addressed the following concerns/questions raised by the reviewers in the individual responses: claim of “scalable model” (reviewer D7TP), lack of fine-grained details (reviewer D7TP, PdSb), use case of our method and extension to dynamic monocular videos (reviewer PdSb, iAZE), concern about novelty and integration of existing components (reviewer veT3, iAZE), lack of sensitivity analysis on loss weights (reviewer iAZE), and feed-forward rendering (reviewer iAZE).

---

> ### Author Response · Authors · 2025-11-26
> **References**
>
> [1] Wang et al., 4D-Dress: A 4d dataset of real-world human clothing with semantic annotations. CVPR 2024.
>
> [2] Wang et al., Dust3r: Geometric 3d vision made easy. CVPR 2024.
>
> [3] Li et al., Pshuman: Photorealistic single-view human reconstruction using cross-scale diffusion. CVPR 2025.
>
> [4] Peng et al., Shape as points: A differentiable poisson solver. NeurIPS 2021.
>
> [5] Han et al., High-fidelity 3d human digitization from single 2k resolution images. CVPR 2023.
>
> [6] Cheng et al., Dna-rendering: A diverse neural actor repository for high-fidelity human-centric rendering. ICCV 2023.
>
> [7] Hu et al., Forge4D: Feed-Forward 4D Human Reconstruction and Interpolation from Uncalibrated Sparse-view Videos. Arxiv 2025.

---

### Meta-Review · Area_Chair_CZt7 · 2026-01-05

**Summary:**

- veT3, iAZE: Questionable novelty, primarily an integration of existing components
- D7TP, PdSb: Geometry quality and fine-grained detail are not sufficiently strong
- D7TP, veT3, iAZE: Limited training data undermines scalability and generalization claims
- D7TP, PdSb, iAZE: Experimental evaluation and ablations are insufficient or unclear

**Reviewer Concerns:**

- The novelty concerns are not well addressed. The rebuttal largely rephrases system integration and component adaptation as novelty, but does not clearly demonstrate conceptual or methodological advances beyond existing techniques.

- While the rebuttal explains the causes of limited fine-grained detail (e.g., grid resolution and memory constraints), concerns about visual fidelity, particularly hands, hair, and high-frequency geometry, remain outstanding and are not substantively mitigated.

- The rebuttal provides new empirical evidence of cross-dataset generalization on DNA-Rendering and 2K2K, and adds a mixed synthetic–real training experiment (THuman + 4D-Dress). These results partially address concerns about training data and generalization. However, the scalability concern is mostly reframed. The rebuttal does not adequately address scalability in terms of data requirements, leaving this issue largely unresolved.

- Additional evaluation and ablation results are provided, addressing the concerns of the reviewers.

**Reviewer Scores:**

As summarized above, while some issues are partially addressed, several major concerns, including novelty, reconstruction quality, and scalability, remain outstanding. Thus, I do not expect any of the reviewers to significantly change their scores if they had been able to participate fully in the discussion.

---

### Decision · Program_Chairs · 2026-01-26

Reject